# Joint Selection for Large-Scale Pre-Training Data via Policy Gradient-based Mask Learning

**Ziqing Fan[1,2], Yuqiao Xian[2,✉], Yan Sun[3], Ke Shen[2], Li Shen[4]**

[1]Shanghai Jiao Tong University; [2]ByteDance Seed; [3]University of Sydney
[4]Shenzhen Campus of Sun Yat-sen University

{fanziqing.knight,xianyuqiao.eric}@bytedance.com, ysun9899@uni.sydney.edu.au,
shenke@bytedance.com,mathshenli@gmail.com

## Abstract

A fine-grained data recipe is crucial for pre-training large language models (LLMs), as it can significantly enhance training efficiency and model performance. One important ingredient in the recipe is to select samples based on scores produced by defined rules, LLM judgment, or statistical information in embeddings, which can be roughly categorized into quality and diversity metrics. Due to the high computational cost when applied to trillion-scale token pre-training datasets such as FineWeb and DCLM, these two or more types of metrics are rarely considered jointly in a single selection process. However, in our empirical study, selecting samples based on quality metrics exhibit severe diminishing returns during long-term pre-training, while selecting on diversity metrics removes too many valuable high-quality samples, both of which limit pre-trained LLMs' capabilities. Therefore, we introduce **DATAMASK**, a novel and efficient joint learning framework designed for large-scale pre-training data selection that can simultaneously optimize multiple types of metrics in a unified process, with this study focusing specifically on quality and diversity metrics. DATAMASK approaches the selection process as a mask learning problem, involving iterative sampling of data masks, computation of policy gradients based on predefined objectives with sampled masks, and updating of mask sampling logits. Through policy gradient-based optimization and various acceleration enhancements, it significantly reduces selection time by **98.9%** compared to greedy algorithm, enabling our study to explore joint learning within trillion-scale tokens. With DATAMASK, we select a subset of about 10% from the 15 trillion-token FineWeb dataset, termed **FineWeb-Mask**. Evaluated across 12 diverse tasks, this high-quality and diverse subset achieves **significant improvements of 3.2% on a 1.5B dense model and 1.9% on a 7B MoE model** after pre-training with hundreds of billions of tokens, demonstrating its effectiveness. Source code is available at: https://github.com/ByteDance-Seed/DATAMASK.

## 1 Introduction

Pre-trained Large Language Models (LLMs) (Yang et al., 2025; Dubey et al., 2024; Seed et al., 2025; Team, 2025; Achiam et al., 2023; Liu et al., 2024) have demonstrated remarkable capabilities, but their pre-training datasets are not publicly available, and there is limited information about their creation. To bridge this gap, many recent innovations have aimed to share their recipes for selecting large-scale data, making pre-training powerful LLMs openly for academic research.

One important ingredient in the recipe is to select samples based on scores produced by heuristic rules, trained LLM judges, or statistical information among text embeddings, which can be roughly divided into quality-aware or diversity-aware metrics. For quality metrics, QuRating (Wettig et al., 2024) and FineWeb-Edu (Penedo et al., 2024) leverage LLMs as judges to

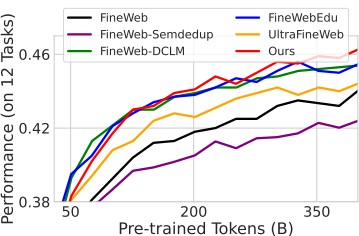

Figure 1: Average performance on 12 tasks across pre-training tokens for a 1.5B dense model trained on FineWeb, FineWeb-Semdedup, FineWeb-Edu, Ultra-FineWeb, FineWeb-DCLM, and our FineWeb-Mask, respectively.

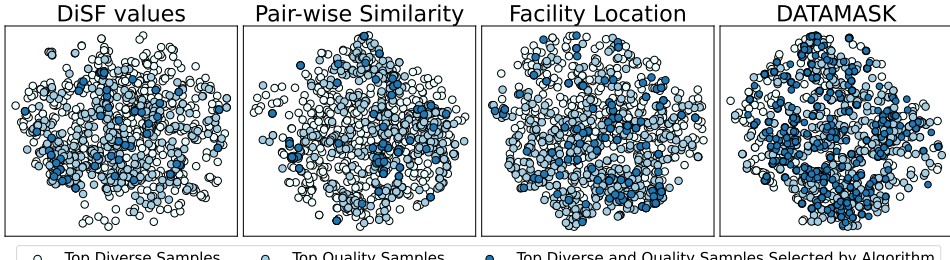

Figure 2: Visualization of text embeddings via t-SNE (Van der Maaten & Hinton, 2008) on random subsets of FineWeb. White, light blue, and dark blue points correspond to samples that are top diverse, top high-quality, and samples selected by algorithms that exhibit both high diversity and quality. Light blue points show tighter clustering. Dark blue points are sparse in algorithms except for ours.

evaluate sample quality (e.g., writing style, educational value, etc.). UltraFineWeb (Wang et al., 2025) and DCLM (Li et al., 2024) employ model based fastText (Joulin et al., 2016) filtering for quality assessment. For diversity metrics, various scores have been proposed to assess coverage or semantic redundancy in machine learning and representation learning. These include pair-wise similarity that measures the average similarity of each text pair in the selected set, facility location (Salhi, 1991a;b; Krause & Golovin, 2014) that evaluates the coverage of the selected set relative to the entire dataset, and DiSF values (Fan et al., 2025) that calculate the dimensional collapse degree of the selected samples' embedding space. Additionally, Semdedup (Abbas et al., 2023), a recent popular method, provides an efficient implementation based on pair-wise cosine similarity that removes samples too close in the embedding space. GneissWeb (Gohari et al., 2025) utilizes sharded exact substring deduplication, followed by an ensemble of quality filters. However, due to the high computational cost when applied to trillion-scale token pre-training datasets such as FineWeb and DCLM, these two or more types of metrics are rarely considered jointly in a single selection process, especially diversity-aware metrics.

In this study, we revisit metric-based selection and observe that selecting samples based on quality metrics shows severe diminishing returns during long-term pre-training, while selecting based on diversity metrics removes too many valuable high-quality samples, both of which limit the capabilities of pre-trained LLMs. As visualized in Figure 2, selecting samples based on quality scores (dark blue and purple data points) leads to tighter clustering compared to the raw data distribution, indicating higher semantic redundancy and reduced information diversity. As purple points shown in Figure 2, when considering only diversity scores, very few valuable high-quality samples are included in the selection. Consequently, as shown in Figure 1, quality-based methods (FineWeb-Edu, Ultra-FineWeb, and FineWeb-DCLM) exhibit promising initial performance but severe diminishing returns when pre-trained on more tokens, while diversity-based methods (FineWeb-Semdedup) demonstrate even poorer performance compared to the raw FineWeb dataset. A straightforward solution is to consider both metrics jointly during single selection. However, unlike quality scores that are calculated individually for each sample, diversity scores are typically defined on a set and its solving with greedy algorithm is often prohibitively costly for trillion-token datasets, making joint learning challenging.

To address this issue, we introduce DATAMASK, a novel and efficient joint learning framework, that conceptualizes the large-scale selection process as a mask learning problem. With a predefined joint objective, it iteratively samples data masks, computes policy gradients according to the objective, and updates the mask sampling logits to converge toward the optimal subset. Through policy gradient-based optimization and acceleration enhancements, the framework achieves a remarkable reduction in selection time by 98.9% compared to greedy algorithm on DiSF, enabling in-depth exploration of joint learning techniques within trillion-token corpora presented in Section 4.3. Under DATAMASK, we produced FineWeb-Mask, a 1.5 trillion token subset of FineWeb, and compare it with multiple large scale open-source corpora by pre-training a 1.5B dense model and a 7B MoE model on hundreds of billions of tokens. Experiments demonstrate strong overall performance improvements on our selected dataset across 12 evaluation tasks, with averaged gains of 2.6% compared to FineWeb, and 0.7% compared to the best baseline. In summary, our contribution can be threefolds:

- We empirically demonstrate the fundamental limitations of single-metric selection on large scale pre-training dataset: quality-only approaches suffer from semantic redundancy, while diversity-only methods suffer from removing too many high-quality samples and high computational costs from greedy algorithm, both limiting the capabilities of pre-trained LLMs (Section 2.3).

- Considering different metric types and the selection time, we propose DATAMASK, a novel and efficient joint learning framework that conceptualizes the selection process as a mask learning problem. Through policy gradient-based optimization and various acceleration enhancements, it significantly reduces 98.9% of selection time compared to greedy algorithms on DiSF, enabling our study to explore joint learning on selecting trillion-scale tokens (Section 2.3 and 4.3).

- Using DATAMASK, we created FineWeb-Mask, a 1.5-trillion-token subset of FineWeb. We compared this subset with other large-scale open-source corpora by pre-training a 1.5B dense model and a 7B MoE model on hundreds of billions of tokens. Evaluated across 12 tasks, we achieved promising improvements in overall performance, with averaged increase of 2.6% compared to FineWeb, and 0.7% compared to the best baseline (Section 4.4).

## 2   RETHINK METRIC BASED DATA SELECTION IN LLM PRE-TRAINING

In this section, we rethink metric-based selection in LLM pre-training, covering the selection objective, related works, and our findings and motivations through empirical analysis.

### 2.1   PROBLEM STATEMENT

Given a complete text set $\mathbb{D} = \{x_i\}_{i=1}^N$, a large language model parameterized by $w$, a pre-training task $\mathcal{L}$, and a performance evaluation tool $\mathcal{A}$, the goal of selection in pre-training is to find the optimal subset $\mathbb{U}^*$ that maximizes $\mathcal{A}$ within a selection budget $S$ and under a predefined token limit $T$:

$$\mathbb{U}^* = \arg\max_{\mathbb{U} \subseteq \mathbb{D}} \mathcal{A}(\arg\min_w \mathcal{L}(w, T, \mathbb{U})), \quad \text{s.t.} \quad |\mathbb{U}| = S. \tag{1}$$

However, directly searching for valuable samples $\mathbb{U}^*$ in the full corpus $\mathbb{D}$ is extremely time-consuming and expensive. To mitigate this cost, recent innovations try to infer the value of samples by defining different metrics. With pre-defined metrics $f(\cdot)$, the selection procedure can be simplified as:

$$\mathbb{U}^* = \arg\max_{\mathbb{U} \subseteq \mathbb{D}} f(\mathbb{U}), \quad \text{s.t.} \quad |\mathbb{U}| = S. \tag{2}$$

### 2.2   RECENT METRIC BASED METHODS

Methods based on scores to select pre-training data can be roughly divided into quality-aware and diversity-aware metrics. Quality metrics are typically derived from heuristic rules, LLM judges, or trained small models using annotated data. For example, QuRating (Wettig et al., 2024) and FineWeb-Edu (Penedo et al., 2024) utilize LLMs as judges to evaluate sample quality in terms of writing quality, educational value, and other criteria. UltraFineWeb (Wang et al., 2025) and DCLM (Li et al., 2024) employ model-based fastText (Joulin et al., 2016) for quality assessment. All these scores are defined on each individual sample, and the values of a selected set can be directly aggregated as

$$\text{Quality Score:} \quad f_{\text{Qua}}(\mathbb{U}) = \frac{1}{S} \sum_{x_i \in \mathbb{U}} Q(x_i), \tag{3}$$

where $Q$ denotes quality scores, including those in FineWeb-Edu, UltraFineWeb and DCLM. For diversity metrics, various scores have been proposed to measure coverage or semantic redundancy in representation learning. These include Pair-wise Similarity that measures the average similarity of each text pair, Facility Location (Salhi, 1991a;b; Krause & Golovin, 2014) that evaluates the coverage of the selected set relative to the entire dataset, and DiSF values (Fan et al., 2025) that calculate the dimensional collapse degree of samples' embedding space, which are mathematically defined as

$$\text{Diversity Score:} \quad f_{\text{Div}}(\mathbb{U}) = \begin{cases} f_{\text{PWS}} = -\frac{1}{2S^2} \sum_{i \in \mathbb{U}} \sum_{j \in \mathbb{U}} \mathbf{K}(z_i, z_j), & \text{(Pair-wise Similarity)} \\ f_{\text{FL}} = \frac{1}{2NS} \sum_{i \in \mathbb{D}} \sum_{j \in \mathbb{U}} \mathbf{K}(z_i, z_j), & \text{(Facility Location)} \\ f_{\text{DiSF}} = -\|\frac{1}{N-1} \sum_{i \in \mathbb{U}} z_i^T z_i\|_F, & \text{(DiSF Values)} \end{cases} \tag{4}$$

where $z_i$ is the embedding of text sample $x_i$, $\mathbf{K}$ is a kernel function used to calculate similarity (cosine similarity in this paper), and $\|\cdot\|_F$ denotes the Frobenius norm. Semdedup (Abbas et al., 2023), a recent popular method, provides an efficient implementation based on pair-wise cosine similarity that removes too close samples in the embedding space. Bigger values of these scores generally indicate a less semantically duplicated and more diverse embedding space. However, due to the high computational cost when applied to trillion-scale token datasets such as FineWeb and DCLM, **these two or more types of metrics are rarely considered jointly in a single selection process**, especially diversity-aware metrics, resulting sub-optimal performance as illustrated in Figure 1.

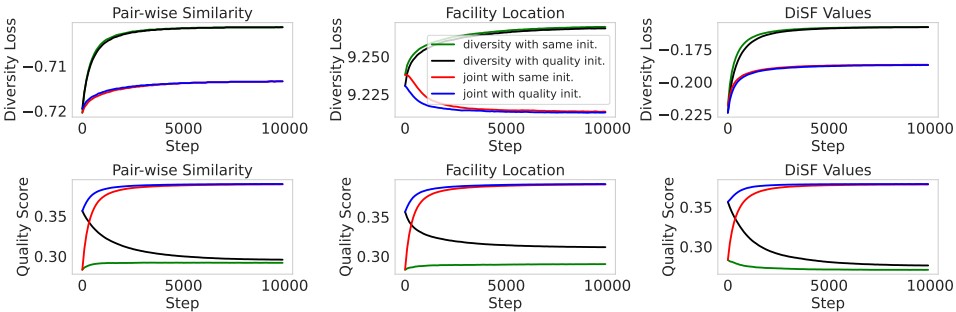

Figure 3: Illustration of the quality and diversity scores when optimizing under DATAMASK with two initialization and two optimization strategies. **Initialization strategies**: same initialization (green and red lines) that all initial sampling probabilities are the same and quality initialization (black and blue lines) that initial sampling probabilities are proportional to the samples' quality score. **Optimization strategies**: diversity optimization (green and black lines) that only optimizes the diversity metric and joint optimization (blue and red lines) that optimizes both the diversity and quality metrics.

## 2.3 MOTIVATION

**Shortcomings of Single Metric-Based Selection.** We first quantify the relationship between diversity and quality metrics under different optimization strategies on FineWeb. When optimizing only diversity scores, there is a sharp decline in quality scores (green and black lines compared to the red and blue lines shown in the bottom three figures of Figure 3). Furthermore, the selected samples show an overly sparse distribution of high-quality samples (purple points visualized in Figure 2). When selection is based solely on quality scores, the embedding space (dark blue points in Figure 2) reveals a tighter clustering of samples. Together, these observations indicate a significant conflict: *quality-based metrics result in higher semantic redundancy and reduced diversity, while diversity-based scores include very few valuable high-quality samples.* Consequenctly, as shown in Figure 1, quality-based methods (FineWeb-Edu, Ultra-FineWeb, and FineWeb-DCLM) demonstrate promising initial performance but suffer from diminishing returns, while diversity-based methods (FineWeb-Semdedup) show even worse performance over the raw FineWeb dataset. In Appendix A.4, we also conducted an experiment in a more semantically duplicated situation, and the model pre-trained on top 20% high-quality samples exhibited worse final performance compared to the raw dataset after long-term pre-training. In both cases, our joint learning approach shows the best performance.

**Dillema on diversity based metrics.** A straightforward approach is to combine the two metrics directly. However, unlike applying the top-$k$ command to select samples based on quality scores defined in Equation 3, diversity scores are typically defined as set functions (Equation 4) and require computationally intensive greedy algorithms to solve. As shown in Figure 4, we present the selection time for a diversity score (DiSF values (Fan et al., 2025)) when selecting 10% of samples from 5,000 to 100,000, compared to our DATAMASK framework when achieving the same optimized value. Results indicate that selection time of greedy algorithm becomes prohibitive when the sample size exceeds 100,000, requiring up to 78 hours, which far exceeds our expectations. With the DATAMASK framework, we significantly reduced the selection time by 98.9%. Although some methods attempt to apply their selection to very small batches (for example, DiSF selects 16 from 1,024), the samples they ultimately select exhibit much poorer diversity scores compared to those optimized on larger batch sizes.

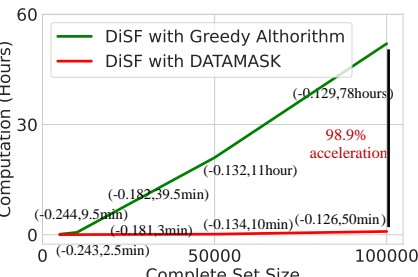

Figure 4: Computation time of DiSF with greedy algorithm and DATAMASK when it reaches the same optimized scores with varying data size. Selection ratio is 10%, and we specifically label the optimized score and required time.

## 3 DATAMASK FRAMEWORK

The shortcomings of single metric-based selection methods, and the computation dilemma of diversity based metrics motivate us to propose following DATAMASK, a novel, effectiveness and efficient joint optimization framework via mask learning used for selecting large scale pre-training data.

**Mask Learning.** In our framework, we adopt a learnable mask $M$ over the full dataset $\mathbb{D} = \{x_i\}_{i=1}^{N}$ and select the expected data subset $\mathbb{U} \subset \mathbb{D}$ as a masking procedure $\Phi_M(\mathbb{D})$:

$$\mathbb{U} = \Phi_M(\mathbb{D}) = \{x_i \in \mathbb{D} \mid M_i = 1\}, \text{ where } |\mathbb{U}| = S, \tag{5}$$

where $M \in \{0, 1\}^N$ is a binary mask vector assigned to each sample. We include the sample $x_i$ in the subset $\mathbb{U}$ whenever the associated indicator $M_i = 1$. Therefore, given a metric $f(\cdot)$ defined w.r.t. samples, the selection can be formulated as an optimization problem of learning an optimal mask:

$$M^* = \arg \max_M \ f(\Phi_M(\mathbb{D})), \text{ s.t. } \sum_{i=1}^{N} M_i = S, \tag{6}$$

where $S$ is the selection budget. The function $f(\cdot)$ can denote quality-based metrics defined in Equation 3, any set function defined in Equation 4, or their combinations.

**Probabilistic Relaxation**. Equation 6 defines the general problem of optimizing a learnable mask $M$. However, its discrete nature of binary values prevents us from utilizing the well-established gradient-based methods to search for the optimal solution $M^*$. In trillion-token-level pre-training datasets, addressing such large-scale combinatorial problems is immensely demanding, posing substantial challenges for optimization. To avoid this problem, we leverage the general probabilistic relaxation of Equation 6 to learn the marginal distribution of the optimal data subset instead of directly optimizing the mask (Hsiao & Sawchuk, 1989; Abbas & Swoboda, 2021; Sun et al., 2025a;b). Specifically, we regard the mask $M$ as being sampled from its optimal distribution $P(M|L)$ w.r.t. the learnable logits $L$. Conditioned on the probability density of selecting each sample, the expected subset $\mathbb{U}$ is obtained via $S$ times without-replacement sampling from the full dataset $\mathbb{D}$. Assuming each mask $M$ is sampled independently w.r.t. logits $L$, we can thus define the density function of $M$ by:

$$P(M|L) = \sum_{\pi \in \text{Perm}(\mathbb{U})} \prod_{k=1}^{S} \frac{P(M_{\pi(k)}|L)}{1 - \sum_{j=1}^{k-1} P(M_{\pi(j)}|L)}, \tag{7}$$

where $\text{Perm}(\mathbb{U})$ denotes the permutation group of $\mathbb{U}$, $\pi(k)$ is the $k$-th element in $\pi$, and the probability of each sample is given by the softmax function: $P(M_i|L) = \frac{e^{L_i}}{\sum_{j=1}^{|\mathbb{D}|} e^{L_j}}$. Thus, the optimization problem becomes that of determining the optimal logits by maximizing the expected scores of $f(\cdot)$:

$$L^* = \arg \max_L \mathbb{E}_{M \sim P(M|L)} \left[ f \left( \mathbb{U} = \Phi_M(\mathbb{D}) \right) \right], \tag{8}$$

In this setting, the learnable parameters $L$ are continuous, which makes them amenable to direct optimization. Yet, because the sampling operation remains discrete, it remains necessary to further enhance the solution procedure to achieve effective optimization.

**Policy Gradient Estimation.** While Equation 8 alleviates the impact of discrete variables, the sampling procedure itself is still non-differentiable. Therefore, we further employ Policy Gradient Estimation (PGE) (Williams, 1992) to accomplish the iterative optimization:

$$\nabla_L \mathbb{E}_{M \sim P(M|L)} \left[ f \left( \Phi_M(\mathbb{D}) \right) \right] = \mathbb{E}_{M \sim P(M|L)} \left[ f \left( \Phi_M(\mathbb{D}) \right) \nabla_L \ln P(M|L) \right]. \tag{9}$$

Generally, we adopt the stochastic form of Equation 9 to estimate the gradient. However, the scale of selected samples through $M$ may be significantly smaller than $\mathbb{D}$, a single estimation can introduce substantial variance and greatly hinder convergence (Xiao & Zhang, 2014). To alleviate this, inspired by the success of relative advantage methods (Guo et al., 2025), we employ a sampling strategy, and calculate the group relative advantage to serve as a baseline for updating, which is,

$$\nabla_L \mathbb{E}_{M \sim P(M|L)} \left[ f \left( \Phi_M(\mathbb{D}) \right) \right] \approx \frac{1}{G} \sum_{j=1}^{G} \frac{f(\Phi_{M^j}(\mathbb{D})) - \mu_G}{\sigma_G} \nabla_L \ln P(M^j|L)), \tag{10}$$

where $M^j$ is $j$-th sampling result following $P(M|L)$, $G$ is the number of sampling times, and $\mu_G$ and $\sigma_G$ are the mean and standard deviation in the group. PGE is widely used in reinforcement learning, and our relative group advantage improve its with lower variance, thereby accelerating training. The detailed theoretical analysis is provided in the Appendix. The update rule is then defined as:

$$L^{t+1} = L^t + \eta \left( \frac{1}{G} \sum_{j=1}^{G} \frac{f(\Phi_{M^j}(\mathbb{D})) - \mu_G}{\sigma_G} \nabla_L \ln P(M^j|L)) \right), \tag{11}$$

where $t$ is the current updating round, and $\eta$ is the pre-defined learning rate.

---

**Algorithm 1** DATAMASK

---

**Input**: Complete set $\mathbb{D}$ (total text samples in this study), selection budget S, objective $f(\cdot)$, learnable logits $L^0$, training epochs E, and group number G.

    **for** $t = 0, \ldots, E - 1$ **do**
        **for** $j = 1, \ldots, G$ in parallel **do**
            Sample the data mask $M^{t,j}$ based on current logits $L^t$ by Equation 7.
            Obtain the selected sample $U^{t,j} = \Phi_M(\mathbb{D})$ by Equation 5, and calculate metric $f(\Phi_M(\mathbb{D}))$.
        **end for**
        Calculate the grouped policy gradient estimator $\nabla_L \mathbb{E}_{M \sim P(L)}\left[f\left(\Phi_M(\mathbb{D})\right)\right]$ as Equation 10.
        Update logits $L^{t+1} = L^t + \eta \nabla_L \mathbb{E}_{M \sim P(L)}\left[f\left(\Phi_M(\mathbb{D})\right)\right])$ as defined in Equation 11.
    **end for**
    Get final mask $M^*$ based on the optimized logits $L^E$ as defined in equation 7.
**Output**: $\mathbb{U}^* = \Phi_{M^*}(\mathbb{D})$

---

**Overall Framework.** Here we restate the overall pipeline of the DATAMASK. As shown in Algorithm 1, we predefine a selection budget S for the text dataset $\mathbb{D}$ and iteratively update the mask logits L for E epochs. In each update epoch, we sample G masks based on current mask logits $L^t$. For each sampled mask, we obtain its corresponding subset of data (Equation 5) and calculate the predefined metrics. With this group of G metrics, we compute the estimated policy gradient (Equation 10), and update the mask logits. Ideally, after E epochs of updates, the mask logits converge to the optimal $L^*$. We then sample the desired data mask $M^*$ from the optimal logits (Equation 7) and obtain the final subset of pre-training data $\mathbb{U}^*$ (Equation 5).

## 4 EXPERIMENTS

In this part, we first introduce the experimental setup, including used dataset, evaluation metrics, model architecture, training details, and baselines in Section 4.1. Then in Section 4.2, we present the metrics used for joint learning. Following this, we provide extensive valuable insights on joint optimization under DATAMASK in Section 4.3. Finally, in Section 4.4, we propose FineWeb-Mask, a high-quality and diverse subset of FineWeb with 1.5 trillion tokens selected with DATAMASK.

### 4.1 EXPERIMENTAL SETUP

**Dataset and evaluation.** FineWeb (Penedo et al., 2024) is a large-scale text corpus derived from 96 Common Crawl snapshots, containing 15 trillion tokens, which is enough to train a Chinchilla-optimal model (Hoffmann et al., 2022) with over 500 billion parameters and proved to produce better-performing LLMs than other open pretraining datasets. Therefore, we decide to explore joint learning with DATAMASK on this text corpus. To comprehensively evaluate the performance of the pre-trained LLMs, we benchmarked across three high-level abilities with twelve diverse tasks, including reading comprehension (RACE-High, and RACE-Middle (Lai et al., 2017)), world knowledge (Trivi-aQA (Joshi et al., 2017), HellaSwag (Zellers et al., 2019), Natural Questions (Kwiatkowski et al., 2019), OpenBookQA (Mihaylov et al., 2018), KQAPro (Cao et al., 2022), and MMLU (Hendrycks et al., 2021)), and reasoning (ARC-Challenge, SIQA (Sap et al., 2019), PIQA (Bisk et al., 2020), and WinoGrande (win, 2019)). Details are provided in Appendix A.3.

**Model architecture and training parameters.** For the model, we use Qwen-2.5-1.5B dense architecture (Yang et al., 2024; Team, 2024). Given the popularity of the Mixture of Experts (MoE) structure, we also explore a 7B MoE model, which consists of 10 experts and has 680M active parameters per token per forward pass. We use E5-base-v2 (Wang et al., 2022) to compute text embedding as defined in Equation 4. During pre-training, we adopt a constant learning rate of 4e-4 with the Adam optimizer, a batch size of 1536, a maximum sequence length of 8192, a weight decay of 0.1, and a gradient clipping threshold of 1.

**Baselines.** To verify the effectiveness of our framework, we compare it to multiple large-scale pre-training datasets specifically on **FineWeb** (Penedo et al., 2024), a 15-trillion-token web dataset. **FineWeb-Edu**, which has 1.3 trillion tokens, was filtered using heuristic methods and a linear regression model built upon the Snowflake-arctic-embed-m embedding model (Merrick et al., 2024).

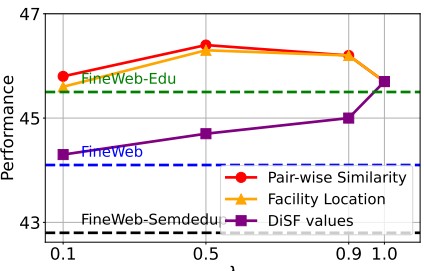

Figure 5: Heatmap of quality score and diversity score in 400 clusters. Deeper color means there are more clusters in a certain range of quality and diversity scores. Analysis are provided in Section 4.3.

UltraFineWeb (Wang et al., 2025) consists of 1 trillion English tokens and 120 billion Chinese tokens, filtered using a lightweight classifier based on fastText (Joulin et al., 2016), and we use its English samples for comparison, named **UltraFineWeb-en**. **FineWebPro** refines a 100-billion-token subset of FineWeb using Prox (Zhou et al., 2024). In addition to FineWeb, we also compare with DCLM-BASELINE (Li et al., 2024) with about 3 trillion tokens, curated from 240 trillion tokens of CommonCraw. Since the raw text sources of DCLM and FineWeb are different, to ensure a fair comparison, we select about 1.5 trillion tokens based on quality score from the DCLM classifier on the FineWeb dataset, naming it **FineWeb-DCLM**. We implemented Semdedup (Abbas et al., 2023) on the FineWeb dataset, naming it **FineWeb-semdedup**. For other diversity metrics (Equation 4), we implemented them within the DATAMASK framework, and conduct ablations in Section 4.3.

## 4.2 DESIGN OF METRICS

As discussed in Section 3, any type of metrics can be jointly leaned with DATAMASK through policy gradient-based optimization. Considering recent works can be roughly categorized into quality and diversity metrics (Section 2), we focus on solving a combination of quality score and diversity score:

$$f(\mathbb{U}) = \lambda f_{qua}(\mathbb{U}) + (1 - \lambda) f_{div}(\mathbb{U}), \tag{12}$$

where $\lambda$ is a tunable parameter balancing quality and diversity. For quality $f_{qua}(\cdot)$, we utilize the sum of DCLM (Li et al., 2024), Edu (Penedo et al., 2024), and Wiki (Touvron et al., 2023a) scores. For diversity $f_{div}(\cdot)$, we conduct an ablation on the metrics defined in Equation 4 in Section 4.3. In Appendix A.5, we provide more details and analysis about how and why we use these scores. Notably, our framework allows for tuning weights between all scores in quality and diversity. To keep the investigation focused on joint optimization between quality and diversity, we concentrate on Equation 12 and leave more complex combinations for further work.

## 4.3 INSIGHTS FROM JOINT LEARNING WITH DATAMASK

**Ablation on diversity metrics and $\lambda$.** As stated in Section 4.2, we decided to select the best diversity metric instead of combining multiple scores, and in Equation 12, we introduce a hyper-parameter $\lambda$ to balance the quality and diversity metrics. Therefore, we conduct ablations on all diversity scores defined in Equation 4 and the balancing factor $\lambda$, by comparing the performance of pre-training a 1.5B LLM with 400B tokens. As shown in Figure 6, pre-trained LLMs benefit from joint learning with Facility Location and Pair-wise Similarity compared to relying solely on the quality score ($\lambda = 1$). Notably, all configurations outperform the raw dataset FineWeb, and a wide range of $\lambda$ values exceed FineWeb-Edu. As for DiSF in joint learning, we observe a negative impact on performance, which may come from its greater inconsistency with quality scores shown in Figure 3. Therefore, in FineWeb, we recommend setting $\lambda$ between 0.1 and 0.5 with Pair-wise Similarity.

Figure 6: Averaged performance when tuning hyper-parameter $\lambda$ defined in Equation 1 compared to FineWeb, FineWeb-Edu, and FineWeb-Semdedup.

**Why diversity scores conflict with the quality score?** To investigate why conflict exists, we cluster all text samples in FedWeb into 10,000 clusters based on their embeddings following K-means algorithm (McQueen, 1967), and visualize the heatmap of diversity scores combined with quality scores. As shown in Figure 5, semantic redundancy exists in both high-

quality and low-quality areas. When directly implementing diversity-based methods, high-quality and low-quality samples are unexpectedly removed equally. With joint learning in DATAMASK, we can control the emphasis on different qualities of samples by adding quality scores as rewards.

**Ablation on choosing** G**.** In Equation 10, we introduce the group number, G. As shown in Figure 7, we record the optimized Facility Location values while tuning G in terms of computational time. Results show that a value of G that is too small causes the training to diverge, while a larger G incurs excessive computational costs. After tuning based on the three diversity metrics, we recommend G = 128 or 256, which is the smallest value that remains stable and yields near-optimal values across all cases.

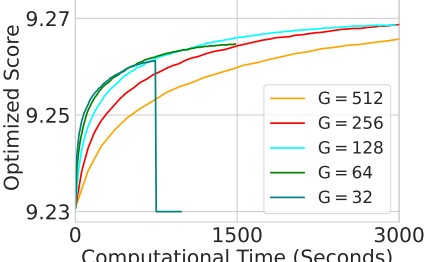

Figure 7: Optimized metrics when tuning group number G in Equation 10.

**Sequence length difference.** During the experiments, we found the averaged selected token length differs significantly. As shown in the Figure 8, we record the token length of selected text documents and comparing them with the raw dataset. It can be seen that, quality based filters like FineWeb-Edu and Ultra-FineWeb prefer longer length of text, which may caused by the bias from the annotated high quality data for training filters. In contrast, diversity based method prefers shorter length of text, since the text embedding is averaged of all tokens, making long length setences has smaller similarity difference.

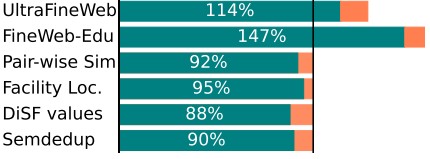

Figure 8: Averaged token length of selected samples compared to the raw dataset. Quality based methods prefer longer documents while diversity based methods prefer to shorter text data.

**Quality-aware pruning and initialization.** When jointly optimized with DATAMASK, we found that the diversity metric can lead to the selection of extremely low-quality data. To avoid this, we filter out the bottom 40-50% of samples based on quality scores. As shown in Table 1, both the selection time and performance are significantly improved. To further mitigate the sampling of extremely low-quality samples, we increase the initial sampling prob-

Table 1: Performance and selection time with DiSF on a 384 GPU platform.

| Method | Performance | Computation |
|---|---|---|
| Random | 44.3 | 18 hours |
| +Pruning | 45.0 | 10 hours |
| +Initialization | 45.1 | 7 hours |

abilities ($L^0$ defined in Algorithm 1) for samples with higher quality score. As red and blue lines shown in Figure 3, the two initialization strategies show similar optimized quality and diversity scores, and the quality-aware initialization accelerates the training process. Except for performance, these findings also provides a way to accelerate the selection procedure under the DATAMASK framework.

**Splitting and batch updating.** When handling text documents in FineWeb, we found that the platform cannot load hundreds of billions of files at once. Therefore, we randomly split the total files into subsets of 1 million samples, which is significantly larger than the 1,024 used in DiSF (Fan et al., 2025). To further reduce computational time, we implemented batch updating, and recorded the converged scores and computational times when randomly selecting a portion of the total samples to update the logits in DATAMASK framework. As shown in Figure 9, batch updating with a ratio of 5% to 10% significantly reduces computation times, and even achieves better converged scores by introducing random noise to escape local optima.

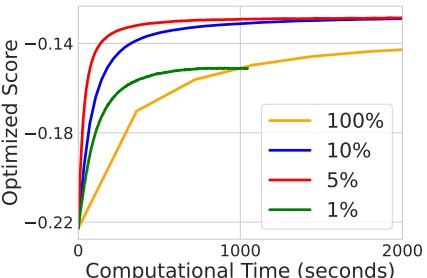

Figure 9: Optimized score and computation when changing the ratio of samples used for mask updating in each step.

### 4.4 FINEWEB-MASK

**Final Strategies.** Here, we select a high-quality and diverse subset of FineWeb using the optimized strategies analyzed in Section 4.3. We first apply quality-aware pruning and initialization, filtering out the bottom $40 - 50\%$ of low-quality samples. Next, for objective in joint optimization, we use quality metric defined in Section 4.2, and Pair-wise Similarity as the diversity metric. By setting

Table 2: The performance on three high-level abilities averaged from 12 tasks. The upper part shows results from a 1.5B dense model pre-trained from scratch with a budget of 400B tokens on each corpus. In the lower part, we show the results of a 7B MoE model pre-trained on 300B tokens.

| Large Scale Corpus | Pre-training 1.4B dense model from scratch with 400B tokens on each corpus | | | | |
| --- | --- | --- | --- | --- | --- |
| | Reading Compre. (2 tasks) | Reasoning (4 tasks) | Knowledge (6 tasks) | Average | Win |
| FineWeb | 46.0 | 53.6 | 38.8 | 44.9 | 0\|12 |
| FineWeb-Semdedup | 46.1 $_{0.1\uparrow}$ | 53.3 $_{0.3\downarrow}$ | 36.7 $_{2.1\downarrow}$ | 43.8 $_{1.1\downarrow}$ | 0\|12 |
| FineWeb-Edu | 46.8 $_{0.8\uparrow}$ | 56.7 $_{3.1\uparrow}$ | 40.2 $_{1.4\uparrow}$ | 46.8 $_{1.4\uparrow}$ | 3\|12 |
| UltraFineWeb-en | 47.4 $_{1.4\downarrow}$ | 56.5 $_{2.9\uparrow}$ | 38.2 $_{0.6\downarrow}$ | 45.8 $_{0.9\uparrow}$ | 0\|12 |
| FineWebPro | 47.6 $_{1.6\uparrow}$ | **57.3** $_{3.7\uparrow}$ | **40.3** $_{1.5\uparrow}$ | 47.2 $_{2.3\uparrow}$ | 2\|12 |
| FineWeb-DCLM | 48.2 $_{2.2\uparrow}$ | 57.2 $_{3.6\uparrow}$ | 39.5 $_{0.7\uparrow}$ | 46.9 $_{2.0\uparrow}$ | 1\|12 |
| FineWeb-Mask (Ours) | **48.8** $_{2.8\uparrow}$ | 56.6 $_{3.0\uparrow}$ | 42.1 $_{3.3\uparrow}$ | **48.1** $_{3.2\uparrow}$ | **6**\|12 |

| Large Scale Corpus | Pre-training 7B MoE model from scratch with 300B tokens on each corpus | | | | |
| --- | --- | --- | --- | --- | --- |
| | Reading Compre. (2 tasks) | Reasoning (4 tasks) | Knowledge (6 tasks) | Average | Win |
| FineWeb | 48.1 | 58.1 | 46.8 | 50.7 | 1\|12 |
| FineWeb-Semdedup | 48.3 $_{0.2\uparrow}$ | 57.6 $_{0.5\downarrow}$ | 45.3 $_{1.5\downarrow}$ | 50.0 $_{0.7\downarrow}$ | 1\|12 |
| FineWeb-Edu | **49.6** $_{1.5\uparrow}$ | 60.5 $_{2.4\uparrow}$ | 46.2 $_{0.6\downarrow}$ | 51.2 $_{0.5\uparrow}$ | 2\|12 |
| UltraFineWeb-en | 48.2 $_{0.1\uparrow}$ | 59.6 $_{1.5\uparrow}$ | 42.7 $_{4.1\downarrow}$ | 49.5 $_{1.2\downarrow}$ | 1\|12 |
| FineWebPro | 49.3 $_{1.2\uparrow}$ | **60.7** $_{2.6\uparrow}$ | 45.9 $_{0.9\downarrow}$ | 51.4 $_{0.7\uparrow}$ | 2\|12 |
| FineWeb-DCLM | 49.0 $_{0.9\uparrow}$ | 60.4 $_{2.3\uparrow}$ | 47.8 $_{1.0\uparrow}$ | 52.2 $_{1.5\uparrow}$ | 1\|12 |
| FineWeb-Mask (Ours) | 49.1 $_{1.0\uparrow}$ | 59.9 $_{1.8\uparrow}$ | **48.9** $_{2.1\uparrow}$ | **52.6** $_{1.9\uparrow}$ | **4**\|12 |

balancing ratio $\lambda = 0.5$, group number G to 128, split size to 1 million, and batch updating ratio to 0.05, updating epochs E to 10,000, we select 1.5 trillion tokens, named FineWeb-Mask, which is comparable in size to FineWeb-Edu. On a platform with 384 GPUs, the selection time is about 15 hours, while pre-training the 1.5B dense model with 400B tokens takes about 2 days.

**Main Results.** We compare FineWeb-Mask with multiple baselines introduced in Section 4.1 on a 1.5B dense model and a larger 7B MoE model. As shown in Table 2, FineWeb-Semdedup that based solely on diversity metric performs even worse than the raw FineWeb on both model types, which may removes too many valuable samples. Quality-based methods like FineWeb-Edu and FineWeb-DCLM outperform raw FineWeb, with gains of 1.4% and 2.0% on the 1.5B dense model, and 0.5% and 1.5% on the 7B MoE model. By using joint learning of quality and diversity metrics with DATAMASK, the performance is further improved, achieving gains of 3.2% and 1.9% compared to FineWeb, and 0.9% and 0.4% compared to the best baseline on the dense and MoE models, respectively.

**Difference in model archetecture.** Notably, as shown in Table 2, different methods exhibit distinct properties on the dense and MoE models. For example, the model pre-trained on UltraFineWeb performs better on the dense model but worse on the MoE model compared to the raw dataset. This differing may be caused by an enlarged gap of reasoning abilities in the dense model, while the gap of knowledge abilities is more pronounced in the MoE model. In both cases, our FineWeb-Mask from DATAMASK framework performs the best.

Selected Data is available at DATAMASK. For performance curves, more experiments, and detailed performance of each tasks, please refer to Appendix A.2, A.4, A.6, A.7, A.8 and A.10.

## 5 CONCLUSION

Due to high computational costs, large-scale data recipes rarely select data based on multiple metrics jointly especially diversity metrics, which shows suboptimal performance during pre-training in our study. Therefore, we propose DATAMASK, a novel and efficient joint learning framework for large-scale pre-training data selection that optimizes multiple metrics simultaneously. DATAMASK treats the data selection process as a mask learning problem, involving iterative data mask sampling, policy gradient computation based on pre-defined objective, and updating sampling probabilities. With policy gradient-based optimization and acceleration enhancements, DATAMASK reduces selection time by at least 98.9% compared to greedy algorithms on DiSF, enabling comprehensive investigation of joint learning with diversity and quality metrics. Using optimized configurations, we introduce FineWeb-Mask, a high-quality and diverse subset of the FineWeb dataset, showing superior performance when pre-training on both the 1.5B dense model and the 7B MoE model.

## 6 ACKNOWLEDGEMENT

The work is done at ByteDance Seed. Li Shen is supported by NSFC Grant (No. 62576364), Shenzhen Basic Research Project (Natural Science Foundation) Basic Research Key Project (NO. JCYJ20241202124430041). Ziqing Fan is partially supported by Wu Wen Jun Honorary Doctoral Scholarship, AI Institute, Shanghai Jiao Tong University.

## 7 ETHICS STATEMENT

This work does not involve any sensitive personal data, human subjects, or animal experiments. The research focuses solely on data selection and large-scale pretraining techniques. All datasets used in this study are publicly available and widely adopted within the research community. We have taken care to ensure that our methods and findings adhere to ethical standards and do not pose foreseeable risks of misuse.

## 8 REPRODUCIBILITY STATEMENT

To ensure the reproducibility of this work, we have provided detailed updating equations and algorithms in Section 3 and Algorithm 1. We also outline the experimental setups in Section 4.1 and present extensive ablations on hyperparameter selection, as discussed in Section 4.3. In Appendix A.2 and Appendix A.10, we include performance curves during pre-training and detailed performance metrics across all 12 evaluation tasks. Furthermore, in Appendix A.11, we provide comprehensive visualizations of text samples with quality scores and similarities for enhanced understanding.

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

# A Appendix

## Contents

## A.1 USE OF LLMS

During the preparation of this submission, large language models (primarily GPT-4o-mini (Hurst et al., 2024), and Gemini-2.5 (Comanici et al., 2025)) were used solely to enhance the clarity and quality of the text, including correcting grammatical and typographical errors, as well as rephrasing sentences for improved readability and fluency. Notably, they were not involved in any substantive work related to the research contributions of this study.

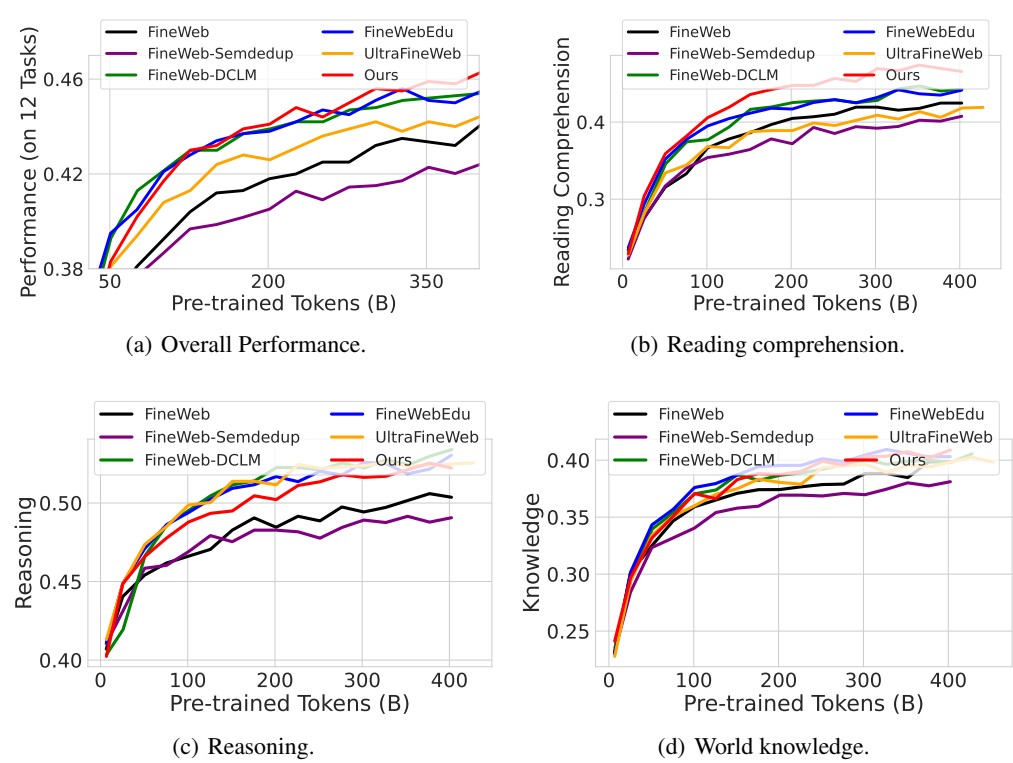

(a) Overall Performance.

(b) Reading comprehension.

(c) Reasoning.

(d) World knowledge.

Figure 10: Performance of overall 12 tasks, reading comprehension, reasoning, and world knowledge when pre-training a 1.5B dense model with 400 billions of tokens.

## A.2 TRAINING CURVES

In this part, we show performance curves during pre-training on the 1.5B dense model and 7B MoE model for better understanding the effectiveness of different methods as shown in Figure 10 and 11. It can be seen that, our selected data show significant improvement compared to FineWeb, especially on the ability of reading comprehension and world knowledge.

## A.3 EVALUATION BENCHMARK

To evaluate the performance of the pre-trained LLMs, we utilize two reading comprehension tasks, four reasoning tasks, and six knowledge tasks, including:

- **TriviaQA** (Joshi et al., 2017): a world knowledge benchmark with closed-book question answering evaluation setting.
- **RACE-High** and **RACE-Middle** (Lai et al., 2017): English examinations in China designed for middle school and high school students.
- **HellaSwag** (Zellers et al., 2019): a collection to assess physically situated commonsense reasoning capabilities.

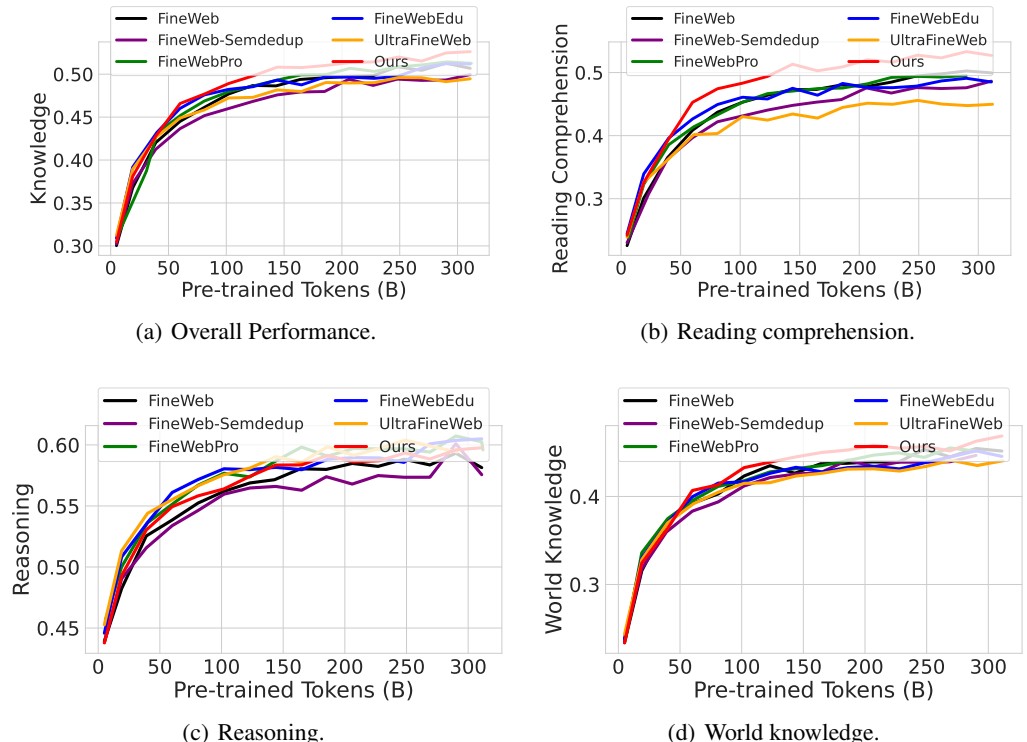

(a) Overall Performance.

(b) Reading comprehension.

(c) Reasoning.

(d) World knowledge.

Figure 11: Performance of overall 12 tasks, reading comprehension, reasoning, and world knowledge when pre-training a 7B MoE model with 300 billions of tokens.

Table 3: The detailed performance on reading comprehension.

| Large Scale Corpus | RACE-High | RACE-Middle | Average |
|---|---|---|---|
| FineWeb | 40.8 | 51.2 | 46.0 |
| FineWeb-Semdedup | 40.5 | 51.7 | 46.1 |
| FineWeb-Edu | 42.3 | 51.4 | 46.8 |
| UltraFineWeb-en | 41.8 | 53.0 | 47.4 |
| FineWebPro | 43.1 | 52.2 | 47.6 |
| FineWeb-DCLM | 43.6 | 52.9 | 48.2 |
| FineWeb-Mask (Ours) | **43.8** | **53.7** | **48.8** |
| Large Scale Corpus | RACE-High | RACE-Middle | Average |
| FineWeb | 42.1 | 54.1 | 48.1 |
| FineWeb-Semdedup | 41.4 | 55.3 | 48.3 |
| FineWeb-Edu | 42.8 | **56.5** | **49.6** |
| UltraFineWeb-en | 42.4 | 54.0 | 48.2 |
| FineWebPro | 42.8 | 55.8 | 49.3 |
| FineWeb-DCLM | **43.2** | 54.8 | 49.0 |
| FineWeb-Mask (Ours) | 42.8 | 55.4 | 49.1 |

- **Natural Questions** (Kwiatkowski et al., 2019): real user questions issued to Google search, with answers found from Wikipedia by annotators.

- **OpenBookQA** (Mihaylov et al., 2018): a collection inspired by open book exams that tests the ability to comprehend and apply knowledge similarly to human understanding.

- **KQAPro** (Cao et al., 2022): complex question answering over knowledge bases (Complex KBQA), which is challenging due to the need for various compositional reasoning capabilities.

Table 4: The detailed performance on world knowledge.

| Large Scale Corpus | HellaSwag | NQ | OBQA | KQAPro | MMLU | TrivalQA | Average |
|---|---|---|---|---|---|---|---|
| FineWeb | 58.9 | 11.1 | 48.6 | 45.1 | 33.9 | 35.2 | 38.8 |
| FineWeb-Semdedup | 57.2 | 11.0 | 45.0 | 43.5 | 33.4 | 30.2 | 36.7 |
| FineWeb-Edu | 57.6 | 12.1 | 53.6 | 42.5 | **37.8** | 37.8 | 40.2 |
| UltraFineWeb-en | 57.8 | 10.9 | 50.6 | 42.2 | 37.2 | 30.6 | 38.2 |
| FineWebPro | 61.3 | 12.4 | **51.5** | 42.0 | 36.2 | 38.6 | 40.3 |
| FineWeb-DCLM | **61.4** | 11.1 | 48.2 | 43.4 | 34.8 | 37.8 | 39.5 |
| FineWeb-Mask (Ours) | 56.4 | **14.1** | 51.4 | **47.0** | 36.5 | **47.3** | **42.1** |
| Large Scale Corpus | HellaSwag | NQ | OBQA | KQAPro | MMLU | TrivalQA | Average |
| FineWeb | **69.9** | 17.9 | 53.6 | 49.8 | 35.8 | 53.8 | 46.8 |
| FineWeb-Semdedup | 67.7 | 17.0 | 53.0 | 49.3 | 35.6 | 49.1 | 45.3 |
| FineWeb-Edu | 65.8 | 16.6 | 53.8 | 48.5 | 41.3 | 50.4 | 46.2 |
| UltraFineWeb-en | 64.9 | 13.9 | **55.4** | 42.0 | **41.4** | 38.6 | 42.7 |
| FineWebPro | 68.9 | 15.5 | 55.2 | 46.3 | 40.0 | 49.8 | 45.9 |
| FineWeb-DCLM | 69.3 | 18.7 | 54.0 | 49.8 | 40.4 | 54.6 | 47.8 |
| FineWeb-Mask (Ours) | 66.1 | **19.5** | 55.0 | **51.4** | 39.5 | **61.7** | **48.9** |

Table 5: The detailed performance on Reasoning.

| Large Scale Corpus | ARC-Challenge | SIQA | PIQA | WinoGrande | Average |
|---|---|---|---|---|---|
| FineWeb | 31.9 | 48.9 | 75.1 | 58.5 | 53.6 |
| FineWeb-Semdedup | 31.6 | 48.7 | 74.9 | 57.9 | 53.3 |
| FineWeb-Edu | **44.5** | 49.2 | 74.2 | 59.0 | 56.7 |
| UltraFineWeb-en | 44.2 | 48.9 | 75.9 | 57.1 | 56.5 |
| FineWebPro | 43.0 | 50.1 | 75.2 | 61.0 | **57.3** |
| FineWeb-DCLM | 40.7 | 50.4 | **76.2** | **61.4** | 57.2 |
| FineWeb-Mask (Ours) | 40.9 | **51.4** | 74.4 | 59.8 | 56.6 |
| Large Scale Corpus | ARC-Challenge | SIQA | PIQA | WinoGrande | Average |
| FineWeb | 40.7 | 52.8 | **78.6** | 63.7 | 58.1 |
| FineWeb-Semdedup | 38.4 | 49.4 | 77.4 | 65.1 | 57.6 |
| FineWeb-Edu | **50.0** | 50.4 | 77.1 | 64.5 | 60.5 |
| UltraFineWeb-en | 49.3 | 49.7 | 78.3 | 60.9 | 59.6 |
| FineWebPro | 47.9 | **52.8** | 77.9 | 64.3 | **60.7** |
| FineWeb-DCLM | 47.0 | 52.1 | 78.5 | 64.0 | 60.4 |
| FineWeb-Mask (Ours) | 45.9 | 51.2 | 77.5 | **65.1** | 59.9 |

- **MMLU** (Hendrycks et al., 2021): a task measuring world knowledge and problem-solving abilities across various subjects.
- **ARC-Challenge** (Clark et al., 2018): a multiple-choice science question set at a grade-school level.
- **SIQA** (Sap et al., 2019): questions about commonsense reasoning in social situations.
- **PIQA** (Bisk et al., 2020): a collection measuring understanding and reasoning about physical interactions in the real world.
- **WinoGrande** (win, 2019): an expansion of the Winograd Schema Challenge (WSC) with increased scale and complexity.

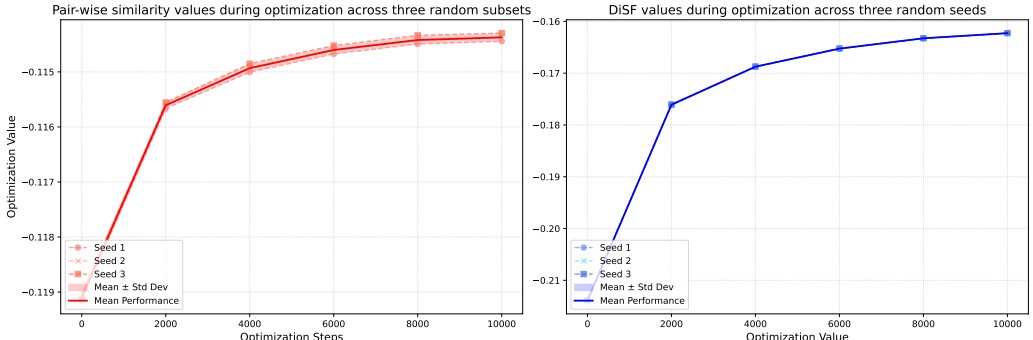

Figure 12: Optimized values and their variance across three different subsets with Pair-wise Similarity, and optimized values and their variance across three different seeds with DiSF values.

## A.4 Experiment on severe semantic deduplication situation

As stated in Section 2.3, we first quantify the relationship between diversity and quality metrics under different optimization strategies on FineWeb. When optimizing only diversity scores, there is a sharp decline in quality scores (green and black lines compared to the red and blue lines shown in the bottom three figures of Figure 3). Furthermore, the selected samples show an overly sparse distribution of high-quality samples (red points visualized in Figure 2). When selection is based solely on quality scores, the embedding space (green points in Figure 2) reveals a tighter clustering of samples. Together, these observations indicate a significant conflict: quality-based metrics result in higher semantic redundancy and reduced diversity, while diversity-based scores include very few valuable high-quality samples. Consequenctly, as shown in Figure 1, quality-based methods (FineWeb-Edu, Ultra-FineWeb, and FineWeb-DCLM) demonstrate promising initial performance but suffer from diminishing returns, while diversity-based methods (FineWeb-Semdedup) show even worse performance over the raw FineWeb dataset. Furthermore, we also conducted an experiment in a more semantically

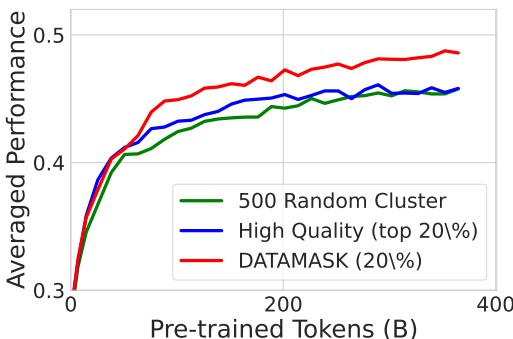

Figure 13: Averaged performance when pre-training a 1.5B dense model with 400 billion tokens on 500 random clusters of FineWeb (approximately 500 billion tokens). We first cluster the FineWeb dataset into 10,000 clusters using the K-means (McQueen, 1967) algorithm and conduct experiments on 500 randomly selected clusters. In these clusters, semantic duplication is more pronounced compared to the entire FineWeb dataset. We compare a sampling of the top 20% high-quality samples with both the raw data and our DATAMASK, maintaining the same sampling budget.

duplicated situation. As shown in Figure 13, we cluster the FineWeb dataset into 10,000 clusters using the K-means (McQueen, 1967) algorithm and conduct experiments on 500 randomly selected clusters. In these clusters, semantic duplication is more pronounced compared to the entire FineWeb dataset. Results show that the model pre-trained on the raw dataset quickly catches up to the top 20% high-quality samples during long-term pre-training, which indicates that while high-quality samples enable a large language model (LLM) to rapidly absorb information from a single source, they reduce the opportunity to learn novel information. *This highlights the necessity for joint learning with diversity metrics, as it allows for the inclusion of samples that might be deemed "not-so-good" by specific quality filters.*

## A.5 Introduction of quality scores

Here we first introduce quality scores used in our quality metric, including:

Table 6: Sample ration of different quality samples in FineWeb dataset.

| Score | 0 | 1 | 2 | 3 | 4 | 5 | 6 | 7 | 8 | 9 | 10 | 11 | 12 | 13 | 14 | 15 |
|---|---|---|---|---|---|---|---|---|---|---|---|---|---|---|---|---|
| Ratio | 3% | 9% | 17% | 24% | 23% | 14% | 6% | 3% | 1% | 0.2% | 0.03% | 0.003% | $\approx$ 0% | $\approx$ 0% | 0% | 0% |

Table 7: The performance on three high-level abilities averaged from 12 tasks after continual pre-training followed by supervised fine-tuning.

| Large Scale Corpus | 7B MoE model | | | | |
|---|---|---|---|---|---|
| | Reading Compre. (2 tasks) | Reasoning (4 tasks) | Knowledge (6 tasks) | Average | Win |
| FineWeb-Edu | 57.4 | 66.2 | 53.0 | 59.1 | 3\|12 |
| FineWeb-Mask (Ours) | **57.5** | **66.4** | **54.4** | **60.2** | **9\|12** |

- **DCLM score** (Li et al., 2024): score produced by a fastText-based (Joulin et al., 2016) filter to obtain high quality samples. The classifier in DCLM is train on about 400k documents split equally between positive and negative classes.
- **Edu score** (Penedo et al., 2024): educational quality scores generated by a trained classifier developed from synthetic annotations from Llama-3-70B-Instruct (Dubey et al., 2024).
- **Wiki similarity** (Touvron et al., 2023b): scores for classifying pages used as references in Wikipedia v.s. randomly sampled pages produced by a trained linear model.

Initially, we aimed to utilize a single quality score as our quality metric. However, all three scores—DCLM score, Edu score, and wiki similarity—are sparse for high-quality samples and indistinguishable in low-quality samples, making data analysis and joint learning challenging. Therefore, we decided to incorporate all three scores and reform the rating system. We first calculate the scores for all the data, then proportionally convert the scores of the other two metrics into a scale of 0-5 based on the distribution of the Edu score. Finally, we sum these three scores to obtain a total score ranging from 0 to 15. As shown in Table 6 and Table 10-16, we illustrate the distribution of sample quality, and visualize sample cases. After manually checking, we found that most samples with a quality score of less than 4 exhibit poor text content. As discussed in Section 4.3, we perform quality-aware pruning by filtering out samples with a quality score lower than 4. This pruning helps joint learning avoid the selection of overly low-quality samples, reduces selection time, and improves the final performance of the pre-trained LLM, as shown in Table 1. Notably, in Figure 6, we conduct an ablation study on balancing the quality metric and the diversity metric. When $\lambda = 1$, the selection relies solely on the three quality scores after quality-based filtering, which show better performance compared to FineWeb-Edu but worse performance compared to our joint learning approach. This illustrates the effectiveness of combining the three quality scores and highlights the further improvements achieved through our joint learning method. As for more complex combinations of these scores, we leave that for future work.

## A.6 MORE ABLATIONS

### A.6.1 ABLATION ON LEARNING RATE

For the learning rate used during joint selection, we conducted a grid search spanning from 0.001 to 1000. As shown in Figure 14, we present the optimization curve based on DiSF value (one type of diversity metric) with different learning rate. From these results, we observed that an excessively

Table 8: Optimization during DVRL compared with our DATAMASK. We record the pair-wise similarity values before optimization, and converged values after optimization, as well as the time to reach the convergence. For DVRL, we also test its performance on unseen set.

| Score | Before opt. | Seen Set (after opt.) | Unseen Set (after opt.) | Computational time |
|---|---|---|---|---|
| DVRL | -0.7101 | -0.6874 | -0.7040 | 2h30min |
| DATAMASK | -0.7101 | -0.6831 | - | 40min |

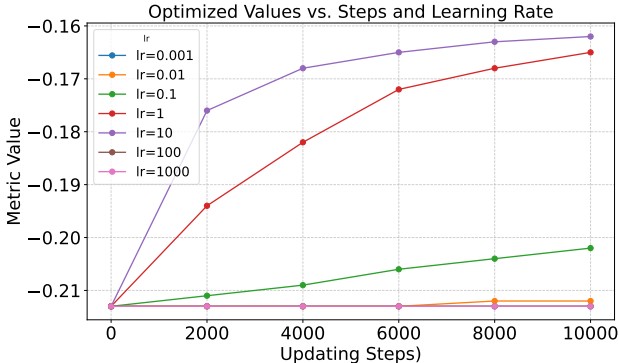

Figure 14: Optimization trajectory with varying learning rate on DiSF values.

large learning rate (e.g., $\geq 100$) led to divergence, while an overly small learning rate(e.g., $\leq 0.1$) hindered the convergence speed. Notably, there is a wide range of learning rates (from 1 to 10) successfully achieve better results than the greedy algorithm with a fast convergence speed. In our experiments, we used a learning rate of 10.

### A.6.2 ABLATION ON INITIALIZATION STRATEGIES

As discussed in Figure 4, and Section 4.3 ("Quality-aware Pruning and Initialization"), we investigate two types of initialization strategies for logits: i) Same Initialization and ii) Quality-Aware Initialization. For same initialization, all logits are all set to 0. For quality aware initialization, logit of i-th sample are set to

$$\text{logit}(x_i) = \frac{f_{\text{qua}}(x_i) - q_{\min}}{q_{\max} - q_{\min}} * (l_{\max} - l_{\min}) + l_{\min}, \tag{13}$$

where $f_{\text{qua}}(x_i)$, $q_{\max}$, and $q_{\min}$ are respectively the quality score, and the maximum and minimum values of the quality score (15 and 0 in our setting). As for determining the maximum and minimum optimized logits $l_{\max}$ and $l_{\min}$, we referenced the largest and lowest logit values observed in the same Initialization setting after convergence. These values were consistently found to be around 5 and -5 in most experimental cases. Figure 3 present the results for the two types of initialization: the final optimized values are similar, while Quality-Aware Initialization significantly reduces the selection time.

### A.6.3 ABLATION ON UPDATE EPOCHS

As illustrated in Figure 3, selecting 10% of the 1 million samples in FineWeb-Mask requires approximately 7,000 to 10,000 update steps. This process takes between 30 minutes and 1 hour on a single GPU, representing a reduction in computational cost of at least 98.9% compared to the classical greedy algorithm. The exact number of update epochs in other settings depends heavily on the user's target optimization values and hyper-parameter configurations. Consequently, most of our ablation studies in Section 4.3—including those on group number, splitting size, pruning, and initialization—aim to minimize the number of epochs required to rapidly reach satisfactory optimized values.

### A.6.4 ABLATION ON SELECTED TEXT LENGTH AND DOMAIN DISTRIBUTION

In this part, we show the distributions of the selected samples' token length and their document domains. As shown in Figure 15, the selected text documents' token length of FineWeb-Mask is between that of the raw FineWeb and FineWeb-Edu. Since the original FineWeb dataset only provides the year of the CC snapshot to which each sample belongs, but not the text domains, we randomly sampled 1,000,000 samples from FineWeb, FineWeb-Edu, and our FineWeb-Mask, respectively. We then used a recently released model TopicClassifier (Wettig et al., 2025) to rate their domains. The categories are ordered by the sample number of FineWeb-Edu. As shown in Figure 16, FineWeb-Edu is skewed towards Science & Tech., Health, and History domains. Conversely, our approach

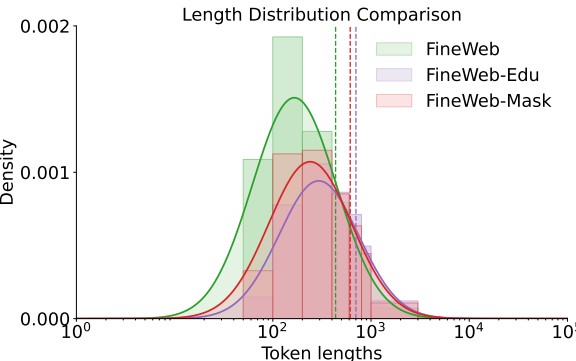

Figure 15: Token length distribution of FineWeb-Mask compared to FineWeb and FineWeb-Edu.

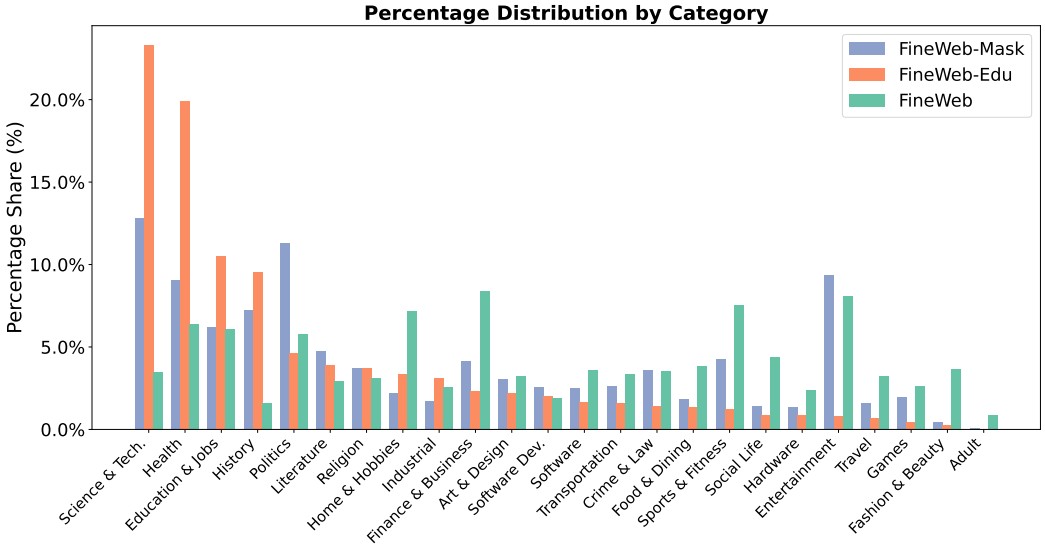

Figure 16: Domain distribution of FineWeb-Mask compared to FineWeb and FineWeb-Edu.

demonstrates the successful trade-off between quality (FineWeb-Edu) and diversity (raw FineWeb distribution).

### A.7 APPLICATION ON CONTINUAL PRE-TRAIN FOLLOWED WITH SUPERVISED FINE-TUNING

We also tested our FineWeb-Mask dataset on the continual pre-training setting, followed by supervised fine-tuning. The base model was initially pre-trained on 400B tokens of FineWeb-Raw data. We then performed the continual pre-training using 200B tokens of FineWeb-Mask and FineWeb-Edu. Then, the subsequent post-training was conducted on recently open sourced dataset: tulu-3-sft-mixture (Lambert et al., 2024). As shown in Table 7, results demonstrate the promising performance of our method and the selected data in continual pre-training followed with supervised fine-tuning. The balance between the educational score and the diversity metric also benefits subsequent post-training.

### A.8 COMPARISON WITH TRADITIONAL LEARNED SELECTION PARADIGMS

Except for us, learned selection paradigms specifically for policy gradient based or model based selection methods have been used in previous methods. For example, DVRL (Yoon et al., 2020) train the data value estimator using a reinforcement signal of the reward obtained from a small

validation set that reflects performance on the target task. DPPNet (Mariet et al., 2019) develop an attention mechanism based on the transformer that captures a notion of dissimilarity between feature vectors. FLEXSUBNET (De & Chakrabarti, 2022) train flexible neural models for both monotone and non-monotone submodular functions for selection. However, different to our approach to training all samples with unique probabilities, these methods typically train a small model to predict the probabilities for a batch of samples. We follow the DVRL pipeline to train a Data Value Estimator, which consists of three linear modules with ReLU activation, followed by a final sigmoid activation. It was trained on 1,000,000 text samples, each with 768 dimensions of features, using a reward based on Pair-wise Similarity (averaged cosine similarity). After convergence, the Data Value Estimator predicted the logits for these 1,000,000 samples as well as for another 1,000,000 unseen samples. Based on these predictions, we selected the top 10% of samples and recorded their exact pairwise similarity compared with our DATAMASK framework. As shown in Table Table 8, DVRL show much worse performance on the converged values than our DATAMASK performance. Unlike our DATAMASK framework, DVRL trains a model to predict the probabilities for a given batch of data, which amplifies the errors arising from the model's learning capacity. Besides, significant larger scores on unseen set compared to seen set indicate the poor generalization ability of DVRL. This means that although DVRL uses a model to predict the probabilities, a new model must be trained when handling new data. Combined with the significant computational demands of a single model (four times the computation compared to our DATAMASK), we may not be able to afford the computation required for large-scale pre-training data. Furthermore, following DVRL's pipeline, we identified significant divergence risks at the 1M sample scale. After careful selection of hyperparameters and engineering efforts , we finally achieved relatively satisfactory optimized values.

## A.9    CONVERGENCE BEHAVIOR AND STABILITY OF DATAMASK

**Empirically**, we demonstrate the optimization trajectory and stability of convergence across three different random seeds and data subsets in this section. As illustrated in Figure 12, the variance of the optimization process across different subsets is minimal, and for different seeds on the same subset, the variance is negligible.

**Theoretically**, we demonstrate the proposed gradient formulation achieves a lower variance than the original policy gradient, which in turn accelerates and improve the stability. We first consider the expectation form of Equation 10, which constructs the average gradient under normalized loss with independently sampled masks. For any component $j$, we have:

$$\mathbb{E}_{P(M|L)}[\mu_G \nabla \log(P(M|L))] = \mu_G \int P(M|L) \frac{\nabla P(M|L)}{P(M|L)} dM = \mu_G \nabla \int P(M|L) dM = \mu_G \nabla 1 = 0.$$

(14)

Therefore, the expectation of its mean is always zero. By denoting the sample loss with respect to the mask $M$ as a function $f(M)$, we have:

$$\mathbb{E}\left[\frac{1}{G}\sum_{j=1}^{G}\left(\frac{f(M_j) - \mu_G}{\sigma_G}\nabla \log(P(M_j|L))\right)\right] = \frac{1}{\sigma_G}\frac{1}{G}\sum_{j=1}^{G}\mathbb{E}_{P(M|L)}\left[f(M_j)\nabla \log(P(M_j|L))\right].$$

(15)

The proposed policy gradient is essentially the mean of the original policy gradients over all samples within the group (differing by a constant factor $\sigma_G$). Therefore, its gradient direction is an unbiased estimate.

Next, we consider the variance. Using the variance expansion formula, we can directly write:

$$\text{Var} = \mathbb{E}\left[\left(\frac{1}{G}\sum_{j=1}^{G}\frac{f(M_j) - \mu_G}{\sigma_G}\nabla \log(P(M_j|L))\right)^2\right] - \left[\mathbb{E}\left(\frac{1}{G}\sum_{j=1}^{G}\frac{f(M_j) - \mu_G}{\sigma_G}\nabla \log(P(M_j|L))\right)\right]^2.$$

(16)

All cross terms vanish because the different masks are sampled independently, and therefore their covariance terms are zero. Then we consider the variance difference between our proposed PGE and vanilla PGE:

$$\Delta \text{Var} = \mathbb{E}\left[\frac{1}{\sigma_G^2}\frac{1}{G^2}\sum_{j=1}^{G}\nabla \log^2(P(M_j|L))\left[(f(M_j) - \mu_G)^2 - f(M_j)^2\right]\right].$$

(17)

It is clear that $\nabla \log^2(P(M|L))$ must be greater than zero. The remaining terms can be further simplified to:

$$(f(M_j) - \mu_G)^2 - f(M_j)^2 = -\mu_G(2f(M_j) - \mu_G), \tag{18}$$

where $\mu_G$ is the average of the values $f(M_j)$ in group $G$. A strict analysis of this formula is challenging because the distribution of $f(M)$ may differ from the distribution of $M$ itself. However, by taking the Gaussian case as an example, we can illustrate the following conclusion. Assume that $f(M)$ follows a normal distribution $\mathcal{N}(\mu_G, \sigma_G)$, then we have:

$$P\left(\frac{1}{G}\sum_i (2f(M_j) - \mu_G) > 0\right) = \Phi\left(\frac{\mu_G\sqrt{G}}{2\sigma_G}\right), \tag{19}$$

where $\Phi$ is the CDF of the Gaussian distribution. Our proposed gradient formulation achieves a lower variance than the original policy gradient with the above probability, which in turn accelerates training. In practice, this probability is very high. By choosing a sufficiently large $G$, we can substantially increase this probability, ultimately ensuring that almost every update step exhibits a variance-reduction effect.

## A.10 DETAILED PERFORMANCE

In this part, we show detailed performance of each task during pre-training on the 1.5B dense model and 7B MoE model as shown in Table 3, 4, and 5.

## A.11 TEXT SAMPLE VISUALIZATION

Here, we present some sample cases for analysis. In Tables 10 to 16, we display samples with quality scores ranging from 0 to 15. It can be observed that samples with a quality score of less than 4 are mostly addresses and advertisements. Additionally, as the quality score increases, the quality of the sentences generally improves, exhibiting a short-long-short phenomenon on text length. In Table 9, we illustrate the nearest and farthest samples compared to a baseline sample using cosine similarity. It can be seen that documents related to "Soundbar" have larger similarity scores, indicating that similarity in feature embedding effectively captures the semantic distance between samples.

Table 9: Closer and farer text samples in embedding space.

**Baseline sample**:

raw: Two removable end speakers take you from 2.1 to 4.1 surround sound
Versatility is key with Philips' Fidelio B5 **Soundbar**. It's a system that includes a main soundbar and wireless sub-woofer, but the soundbar includes two end speakers that can be detached and situated as rear speakers in order to supply surround sound.
Essentially, it's a 4.1-channel system when the speakers are detached from the soundbar, and 2.1 when the speakers are attached to it. The end speakers work wirelessly, with batteries that can last for five hours in music mode, and 10 hours in movie mode.
The detached speakers and the soundbar can also form part of a multi-room music system using their proprietary wireless technology, or be used as portable, independent Bluetooth speakers. It's these features that allow this multi-speaker system to be useful for multiple scenarios.
Specs include a 6.5in woofer driver for the sub-woofer; there are two 3in woofers and two 1in dome tweeters in the main unit; the portable speakers feature 3in full-range drivers. Total power output is 210W. Connections include a 3.5mm port for an MP3 player, analogue (left/right) input, coaxial input, optical input, two **HDMI** inputs, one HDMI out, and USB (for software upgrades).
Don't have an account? Sign up here

**Closer sample (Pair-wise similarity=0.9375, both soundbar)**:

New Delhi (India), June 28: To cater to the growing proliferation of smaller TVs and the huge increase in OTT consumption, Amkette – one of India's leading brands in consumer electronics, today announced the launch of Amkette AMP Audacity 1000 Digital Soundbar in the home audio category, exclusively available on Amkette Store, Amazon, and Flipkart.
The new Audacity 1000 **Soundbar** delivers a dynamic and immersive sound output and comes with the latest advanced features one can expect on higher-priced options. The Amkette AMP Audacity 1000 is sleek and beautiful and is the ideal size for 43 and smaller televisions. The 40 Watt output powered by 2 large 50mm speakers is perfectly tuned for small to mid-sized rooms and it comes with multiple connectivity options, including HDMI input which is unique at this price range. The Amkette AMP Audacity 1000 is priced at 4999 and will be available at an exclusive limited-time launch price of INR 2999 on ... The Sleek and Shiny finish complements any TV and will exquisitely blend into the aesthetics of the room. **HDMI** Input for Better Audio and Huge Convenience The Amp Audacity 1000 is the only 40W Soundbar with HDMI ARC input. HDMI ARC is the only way to experience truly immersive sound because it can deliver richer sound than the normal AUX output. Not only can you experience a rich bass but also bless your ears with soothing music with rich details. But that's not all – with the Soundbar connected to an HDMI ARC port on the TV, using the provided HDMI cable –one can use the single TV Remote to control the soundbar as well. Simply turning the TV on, will turn the Soundbar on and more. Once HDMI Arc is experienced no one would prefer to go back.
Multiple Connectivity Options

**Farer sample (Pair-wise similarity=):0.744**

Being here means that you are looking for one of the most updated home theater systems in the market! Note that you chose perfectly in entering our website! A really great selection of such type of objects is waiting to become yours!
Step forward now! Discover below a nice list of kardon systems. Feel free to click on each product, in order to get further infos about their features! Compare them and spot your next home theater system immediately!
A unique movie experience is here! Live it now!

Table 10: Sample visualization with different quality scores (part 1).

**Sample 1 (Quality score=0)**:

This web page or document may have been moved, deleted or had its name changed. Please use the links below to access some of our most popular web pages, search our site using our site map or use our search tool.
Make an Online Payment
Manage Your NJM NJ Auto Insurance Policy
Auto Insurance Information
New Jersey Auto Insurance Quotes
Pennsylvania Auto Insurance Quotes

**Sample 2 (Quality score=1))**:

Williams Fine ImagesPhotographer Male Plattsburgh, New York, US
My Website: William's Fine ImagesMy MM URL:
http://www.modelmayhem.com/williamsfineimages
Mayhem # 1306161
Other places you can find me...
FB.. Williams-Fine-Images-Retired Photographer please check out and "like"
Experimental, Travel and Fetish images
I am a retired photographer now enjoying the freedom to experiment with images that are not your normal, "oh look, how nice" style. Adults is who I want to see my images, models with an open mind is who I work with. Come to the dark side of my visual mind.
Due to illness, ONLY personal projects will be shot going forward.
ONLY contact me if you are very open minded please. I will no longer shoot dull uninteresting images.
*Yes, please bring a second person so we can both feel safe about what we decide to do.
* Your ideas always read and listened too.
* I always ASK; you get to say yes or no to concepts and poses and everything else.
* You get edited files to use as you wish, if you travel, money to cover at least most of your expenses and a model release is expected and payment to you made .
FYI, the BEST list of art images on MM, hands down.
Make up artist MM#857032 ****
AngelDawn Modeling #2173
Modern Edges makeup and hair #2788036
and a lot more over the years not on MM

**Sample 3 (Quality score=2))**:

to create and rate content, and to follow, bookmark, and share content with other members.
FPGA timing with ATC2 cores
Question asked by
on Jan 19, 2007
on Jan 26, 2007 by Brig Asay
Show 0 Likes
One other question. What timing impact should I anticipate using Agilent's ATC2 core with a Spartan 3 device? Also, should I plan on removing the core when I ship or should I leave it in?
......

Table 11: Sample visualization with different quality scores (part 2).

---

**Sample 4 (Quality score=3)):**

To search for beekeepers in Ohio who will remove swarms or established hives click on your county below:
- Check and make sure they are honey bees. Yellow jackets and wasps are often mistaken for honey bees. See this identification guide for more info.
- Once a swarm moves into a wall or hollow tree it is no longer a swarm and may need cut out. Many beekeepers do not provide this service due to the difficulty and expertise needed. Look for hive removal or cutout in the search results for beekeepers who will.
- Don't expect a beekeeper to remove a swarm for free. A few will depending on location, but free bees often costs more time and gas than purchasing bees. Hive removal and cutouts are rarely free due to the time, expense and liability involved.
- Beekeepers and companies listed here are not endorsed by the Ohio State Beekeepers Association. They are listed here for information only.
- Do you remove swarms or established hives? You must be a current member of OSBA to be included on the list. Join OSBA or update your membership information to be included on the swarm list.
Swarm removal in Mahoning County (Search another county.)
Tim Cassidy (Mahoning County)
Call or text 330 540 3211 Youngstown and surrounding areas within 10 miles.
Swarm removal in counties adjacent to Mahoning County
Donald Kehl (Stark County)
swarm removal and cutouts removed. swarm removal is free, prices for cutouts ( established hives in walls) also free 15 miles radius of louisville,ohio 330 875 4066
...
**Sample 5 (Quality score=4)**:

Join or Renew and Choose Your Gift
- Offer ends Dec. 17
- Discounts on travel and everyday savings
- Subscription to AARP The Magazine
- Free membership for your spouse or partner
A master at dissecting politics through humor, The Daily Show host claims to "represent the distracted center." In the past few months he won his 10th consecutive Emmy, and debated Bill O'Reilly live in what was billed as "The Rumble in the Air-Conditioned Auditorium."
Jamie McCarthy/Getty Images for The Rumble
A victim of a tabloid tizzy over her divorce from third husband Ashton Kutcher, the actress is still picked over by the media — most recently for being "sad and thin." May she live forever in our memories as the ingenue in the pottery-wheel scene with Patrick Swayze in Ghost.
Joe Stevens/Retna Ltd/Corbis
The former Brat Pack actor of Pretty in Pink fame is now — who knew? — a travel writer, working as an editor-at-large at National Geographic Traveler. He's also just come out with a memoir, The Longest Way Home.
Matt Carr/Getty Images
The comedian with a hit sitcom 20 years ago is now serious about politics: This year she ran as the presidential nominee for the Peace and Freedom Party. She lives with her husband on a Hawaiian macadamia nut farm.
Mark Davis/WireImage/Getty Imags
... Join Today

Table 12: Sample visualization with different quality scores (part 3).

**Sample 6 (Quality score=5)):**

19th December, 1947

I am glad to note that the overwhelming majority of the leading nations in the world should have recognized the claim of the Jewish People to establish an Independent Jewish state, in Palestine and should have promised armed assistance to get it realized. After centuries of sufferings, sacrifices and struggle the Jews will soon recover their national Home in Palestine which has undoubtedly been their Fatherland and Holy-land. Well may they compare this event to that glorious day in their history when Moses led them out of the Egyptian bondage and wilderness and the promised land flowing with milk and honey came well within sight. Judging from the Indian Press in general our public seems to be misinformed by a sinister pro-Moslem propaganda regarding this Palestine issue. It must be emphasized therefore that speaking historically, the whole of Palestine has been, from at least two thousand years before the birth of the Moslem Prophet, the National Home of the Jewish people. A long line of their great prophets and kings, of Abraham and Moses, of David and Salomon, has endeared that country to them as their Fatherland and Holy-land. The Arabian Moslems invaded Palestine only a few decades before they invaded our Sindh and just as their fanatical fury exterminated the ancient Egyptians or Persians, they attempted to wipe out with fire and sword the Jewish people too. But they failed in this unholy ambition. The Fatherland or the Holy-land of the Arabian Moslems lies in Arabia and not in Palestine.

In justice, therefore, the whole of Palestine ought to have been restored to the Jews. But taking into consideration the conflict of self-interests of the powerful nations in the UNO, their support to the resuscitation of the Jewish State in a part of Palestine at any rate wherein they still happen to be in majority and which includes some of their prominent Holy Places constitutes an event of historical justice and importance.

...

**Sample 7 (Quality score=6)):**

Of all presidential reputations, Andrew Jackson's is perhaps the most difficult to summarize or explain. Most Americans recognize his name, though most probably know him (in the words of a famous song) as the general who "fought the bloody British in the town of New Orleans" in 1815 rather than as a two-term president of the United States from 1829 to 1837.

AP US History Document Based Question Directions: The following question requires you to construct a coherent essay that integrates your interpretation of Documents A-I and your knowledge of the period referred to in the question. High scores will be earned only by essays that both cite key pieces of evidence from the documents and draw on outside knowledge of the period. Jacksonian Democrats.

Andrew Jackson (Democrat) ran for re-election with V.P. Martin Van Buren. The main issue was his veto of the recharter of the U.S. Bank, which he said was a monopoly. Henry Clay (Whig), who was pro-Bank, ran against him The Anti-Masonic Party nominated William Wirt. This was the first election with a national nominating convention. Jackson won - 219 to Clay's 49 and Wirt's 1. The Masons were a.AP US History DBQ ESSAY Throughout the period dating from 1801 to 1817, the United States government was primarily controlled by the Jeffersonian Republican party, whereas the Federalist Party began to slowly fade away from public view. The Jeffersonian Republican party, led by Thomas Jefferson, professed to favor a weak central government.Andrew Jackson was born to Presbyterian Scots-Irish immigrants Andrew and Elizabeth Jackson, on March 15, 1767 approximately two years after they had emigrated from Carrickfergus. (2)(3) Three weeks after his father's death, Andrew was born in the Waxhaws area near the border between North and South Carolina. He was the youngest of the Jacksons' three sons. His exact birth site was the.

...

Table 13: Sample visualization with different quality scores (part 4).

**Sample 8 (Quality score=7)):**

Leprosy has been eradicated in Culion since the 1980s but the stigma still remains. It didn't help that World Health Organization (WHO) only declared Culion leprosy-free in 2006. So why not embrace this history of healing of this infectious disease? The structures, the research, the tools and the story of the people who lived while it was a leper colony still remains. Culion town is a witness on how a community persevered and healed.
La Immaculada Concepcion Church
The Culion Church of La Immaculada Concepcion Church is an eye-catching structure on a hill. It's hard to miss its bright red wall facing the sea when approaching the town. I had written about the story of this church when I visited in 2013. The charming Hotel Maya beside it has become a quarantine facility for COVID-19. The back of the church where the seal, canon and watchtower has been cordoned off at the time of my visit. This is to prevent people from congregating in the area. Nothing much has changed with the church itself which is a good thing.
Loyola College of Culion
Just beside the foot of the stairway leading to Culion Church is another notable heritage structure, the Loyola College of Culion. The school was established in 1936 by the Jesuits as Culion Catholic Primary School. It was an exclusive school for the children of the leper patients. But by mid-1950s with the enactment of the Liberalization Law for Lepers, the doors of the school were opened to everyone. The school changed its name several times and settled to Loyola College of Culion in 1988.
Culion Museum and Archives
The Culion museum is the heart of Culion's heritage. A two story building within the Culion Sanitarium houses the Culion Leprosy Archives. Thanks to the initiatives of the Philippine National Commission for UNESCO, this documentary heritage was officially inscribed to the UNESCO Memory of the World Register – Asia and the Pacific Region on June 18, 2018. A step closer to be a part of the international list and be recognized as a World Heritage Site.

**Sample 9 (Quality score=8):**

Round London Sightseeing Tour
- Sheet: 39 x 25 inches (99.1 x 63.5 cm)
In artist's hand, in black, bottom right corner: "A. GAMES". In black, in underground symbol at bottom, center: "ROUND LONDON— Over 20 miles of City and West End. Daily, hourly.— 10 00 - 21 00. Lasts about 2 hours. No booking.— From Piccadilly Circus and Buckingham Palace Rd— (near Victoria Station). FARE 50p (Child 30p).— SIGHTSEEING TOUR."
Lettered, lower center: "ROUND LONDON — Over 20 miles of City and West End. Daily, hourly, 1000-2100. Lasts about 2 hours. No booking. From Piccadilly Circus and Buckingham Palace Rd (near Victoria Station). FARE 50p (Child 30p). — SIGHTSEEING TOUR"
In black, bottom right corner: "A. GAMES"
- Credit Line:
...

Table 14: Sample visualization with different quality scores (part 5).

**Sample 10 (Quality score=9)):**

Potentially doubling US offshore wind capacityMassachusetts Deval Patrick and U.S. Interior Secretary Sally Jewell announced plans for a new proposed offshore wind power area of more than 742,000 acres, or 1,160 square miles, which would make it about the size of Rhode Island (1,214 sq-miles). This new area, where space would be auctioned in 4 different leases, would nearly double the federal offshore acreage available for large wind energy projects.

Secretary Jewell said that the government has learned from the Cape Wind offshore wind project in Nantucket Sound, which faced over a decade of opposition and lawsuits, and have picked a spot farther from the shore that should not be as contentious."We put in zones that we believe have both high potential and lower conflict," Jewell said. "But it's going to actually get down to a specific construction plan on a specific site and (an environmental) analysis to determine what people want to do economically and what that impact is going to be.

The edge of the area would be 12 miles off Martha's Vineyard and 13 miles off Nantucket, but turbines wouldn't be erected there, only farther in the zone, which is significantly farther away than one can see because of the curvature of the Earth. For example, an observer that is 5 feet 7 inches tall can see about 2.9 miles away if there are no obstacles in the way. Even if you were high enough in the air to see 12 miles away, everything there would be incredibly small. Add to that refraction from the air and distortion from humidity, etc. So if the turbines are very tall and the day is very clear, they might be visible, maybe. Probably not.

Of course, that probably won't stop the NIMBY people, but the electricity that we use has to come from somewhere; better from the wind than from carbon buried in the Earth's crust...

...

**Sample 11 (Quality score=10)**:

NASA Artist Impression of Kepler-37b

—Discovery date——February 20, 2013—

Kepler-37b is an extrasolar planet (exoplanet) orbiting Kepler-37 in the constellation Lyra. As of February 2013[update] it is the smallest planet discovered around a main-sequence star, with a radius slightly greater than that of the Moon. The measurements do not constrain its mass, but masses above a few times that of the Moon give unphysically high densities.

Kepler-37b, along with two other planets, Kepler-37c and Kepler-37d, were discovered by the Kepler space telescope, which observes stellar transits. After observing transits of Kepler-37b, astronomers had to compare it with the size of the parent star.

The size of the star was obtained using asteroseismology;[clarification needed] Kepler-37 is currently the smallest star to be studied using this process. This allowed the size of Kepler-37b to be determined "with extreme accuracy".

To date, Kepler-37b is the smallest planet discovered around a main-sequence star[b] outside the Solar System. Detection of Kepler-37b was possible due to its short orbital period, relative brightness, and low activity of its host star, allowing brightness data to average out quickly. The discovery of Kepler-37b has led Jack Lissauer, a scientist at NASA's Ames Research Center, to conjecture that "such little planets are common".

Kepler-37b is located approximately 210 light-years from Earth. It is slightly larger than the Moon, with a diameter of about 3,900 kilometres (2,400 mi). NASA states that it probably has no atmosphere and cannot support life. Furthermore, it is most likely composed of rocky materials. Because it is so close to its star (Mercury is more than three times as far from the Sun), Kepler-37b's mean temperature is estimated to be around 425 °C (800 °F).

...

Table 15: Sample visualization with different quality scores (part 6).

**Sample 12 (Quality score=11)):**

Researchers find sex bias in the natural history collections of museums around the world

Researchers have found an unlikely but compelling example of sex bias in the natural history collections of museums around the world.Posted — Updated

The fact is that visitors are more likely to find male than female specimens in the Natural History Museum in London or the Smithsonian in Washington.

atalie Cooper, a researcher from the museum in London, and her colleagues looked at almost 2.5 million specimens from five international collections and concluded that there was a bias towards male specimens. in particular, 40% of the bird specimens and 48% of the mammals analyzed were females.

"We suspected we'd see a bias towards males because science is done by people and people have an inherent male bias," Cooper told CNN via e-mail. "This is especially true as many of our collections come from the Victorian era of macho hunters going out trying to shoot the biggest fiercest creatures for their collections."

More interesting, perhaps, is that the proportion of female specimens hasn't really changed in the past 130 years.

"We were quite surprised by this as we thought things would be getting better," Cooper said. "It could be unconscious bias where people don't even realize they're selecting males, it could be passive in that males are easier to catch or easier to see in the wild, or maybe conservation concerns leading collectors to avoid females."

Cooper said unconscious bias probably played a big role, as both men and women tend to be biased towards males.

Collection methods need also to be considered as a factor for the bias. Males can be showier or larger in the wild, thus easier to collect.

"One good example is the way we usually collect birds. You put up a mist net, then play male bird calls to attract other birds," Cooper said.

...

**Sample 13 (Quality score=12):**

Wade–Giles, sometimes abbreviated Wade, is a romanization system for Mandarin Chinese. It developed from a system produced by Thomas Wade, during the mid-19th century, and was given completed form with Herbert Giles's Chinese–English Dictionary of 1892.

Wade–Giles was the system of transcription in the English-speaking world for most of the 20th century, used in standard reference books and in English language books published before 1979. It replaced the Nanjing-based romanization systems that had been common until late in the 19th century, such as the Postal romanization (still used in some place-names). In mainland China it has been entirely replaced by the pinyin system approved in 1958. Outside mainland China, it has mostly been replaced by pinyin. Additionally, its usage can still be seen in the common English names of certain individuals and locations such as Chiang Ching-kuo.

Wade–Giles was developed by Thomas Francis Wade, a scholar of Chinese and a British ambassador in China who was the first professor of Chinese at Cambridge University. Wade published in 1867 the first textbook on the Beijing dialect of Mandarin in English, the Yü-yen tzu-erh chi, which became the basis for the Romanization system later known as Wade–Giles. The system, designed to transcribe Chinese terms for Chinese specialists, was further refined in 1912 by Herbert Allen Giles, a British diplomat in China and his son, Lionel Giles, a curator at the British Museum.

Table 16: Sample visualization with different quality scores (part 7).

**Sample 14 (Quality score=13)):**

The diagram shows how electricity is generated by a hydroelectric dam.Write a 150-word report for a university lecturer explaining how the process works.
The diagram illustrates the basic principles of hydroelectric power. The process requires the construction of a large dam connected to a powerhouse. The dam creates a large reservoir and the powerhouse is where the electricity is generated.
First of all, water trapped in the reservoir behind the dam is forced through an intake. It then flows into a narrow chamber called a penstock, where the resulting high pressure turns a turbine. The turbine is connected to a generator in the powerhouse above, and this is where the movement of the turbine is converted into electricity. The resulting electricity leaves the powerhouse via cables that carry it over long distances to where it can be used.
It is interesting to note that a hydroelectric dam creates no harmful byproducts and relies entirely on natural forces to produce electricity. After the turbine stage, water flows out through a second channel and into a river. The process is renewable, thanks to the water cycle in nature.

**No sample has quality score=14**

**No sample has quality score=15**

