# OpenReview forum: "Joint Selection for Large-Scale Pre-Training Data via Policy Gradient-based Mask Learning"
_ICLR.cc/2026/Conference — ICLR 2026 Poster_

### Official Review · Reviewer_NCCX · 2025-10-15

**Soundness:** 3
**Presentation:** 3
**Contribution:** 3
**Rating:** 6
**Confidence:** 3

**Summary:**

This paper introduces DATAMASK, an efficient framework for large-scale pre-training data selection by jointly optimizing for both quality and diversity. DATAMASK reframes data selection as a mask learning problem, using a policy gradient-based approach to learn an optimal mask distribution. Instead of exhaustive comparisons, it iteratively samples small data batches, computes joint metrics, and updates a global sampling probability for each data point. By applying DATAMASK to the 15 trillion-token FineWeb dataset, the authors created FineWeb-Mask, a high-quality and diverse 10% subset, which achieves significant improvements over baselines trained on the random subset.

**Strengths:**

1. The proposed DATAMASK algorithm is highly efficient and capable of handling data selection on large-scale datasets.
2. Based on pre-training results, the resulting data subset demonstrates higher quality than other existing subsets.

**Weaknesses:**

1. The necessity of the policy gradient algorithm is questionable. A simpler approach, such as directly assigning logit values based on each sample's quality and diversity score (evaluated on a small group), might achieve a similar effect.
2. The assumption that sample selection probabilities are independent is a potential limitation. This modeling choice seems inconsistent with the goal of optimizing for diversity, which is inherently a set-level property that depends on the relationships between samples.
3. The algorithm's performance is likely sensitive to the learning rate, yet a corresponding analysis is absent. A low learning rate risks insufficient differentiation among logits, approximating uniform sampling, whereas a high learning rate may lead to deterministic, greedy-like behavior. The effect of this hyperparameter on model performance remains unclear.
4. The writing in some sections is not clear enough, for example, the explanation of the content in Figure 3.

**Questions:**

None

---

> ### Author Response · Authors · 2025-11-21
> **Rebuttal to Reviewer NCCX (1/2)**
>
> **Thanks for the constructive feedback provided by the Reviewer NCCX. We sincerely appreciate the time and effort you dedicated to evaluating our work. Below, we provide detailed responses to the weaknesses and questions, which we denote as W and Q. Besides, we also added the following discussions into revised submission.**
>
> > **W1:** The necessity of the policy gradient algorithm is questionable. A simpler approach, such as directly assigning logit values based on each sample's quality and diversity score (evaluated on a small group), might achieve a similar effect.
>
> **Reply to W1:**
>
> Directly assigning logit values based on evaluating the diversity score on small groups faces two main issues:
> - **Unstable and Inaccurate Logit Estimation.** Since the estimation of diversity score is performed in a small group and cannot interact with other groups, the resulting estimation may be unstable and inaccurate. As shown in Figure 7 of the original submission, **even averaging 64 estimation still led to optimization divergence** in an update step. This indirectly demonstrates highly inaccurate and unstable logit estimation of directly assigning logit values.
> - **Same Diversity Scores within the Group.** As diversity score defined in Equation 4, all samples estimated within a group have the same diversity value. This shifts the selection target from selecting individual samples to selecting groups, which is not what we expected. Notably, all metrics based on set functions (not only the diversity metric) face this same issue.
>
> Thinking further, assigning logits based on multi-turn estimation may solve the two issues, by averaging each sample's diversity score across different sampling groups. However, it follows a similar spirit and procedure to the rollout sampling and updating steps in our learning algorithm.
>
>
> > **W2:** The assumption that sample selection probabilities are independent is a potential limitation. This modeling choice seems inconsistent with the goal of optimizing for diversity, which is inherently a set-level property that depends on the relationships between samples.
>
> **Reply to W2:**
>
> **We apologize for the potential misunderstanding of this statement. We intended to express that the rollout procedure itself is independent, rather than the sampling of each individual sample is independent.** Based on Equation 7, the sampling process induced by our assumed distribution is not independent across samples. We have revised this statement in the new submission.
>
>
> > **W3:** The algorithm's performance is likely sensitive to the learning rate, yet a corresponding analysis is absent. A low learning rate risks insufficient differentiation among logits, approximating uniform sampling, whereas a high learning rate may lead to deterministic, greedy-like behavior. The effect of this hyperparameter on model performance remains unclear.
>
> **Reply to W3:**
>
> We thank the reviewer for their constructive comments. Here, we provide details regarding the learning rate. During joint selection, we conducted a grid search spanning from 0.001 to 1000 on a spllited batch (selecting 10% samples from about 1,000,000 documents). The following table presents the results based on DiSF values (one type of diversity metric). From these results, we observed that an excessively large learning rate (e.g., >=100) led to divergence, while an overly small learning rate(e.g., <=0.1) hindered the convergence speed. **Notably, there is a wide range of learning rates (from 1 to 10) converged to a similar satisfactory value with a fast convergence speed.** In our experiments, we used a learning rate of 10. This discussion has been added to Appendix A6.1 of the revised submission.
>
> | | 0 step | 2000 step | 4000 step | 6000 step | 8000 step | 10000 step |
> | :--- | :--- | :--- | :--- | :--- | :--- | :--- |
> | lr=0.001 | -0.213 | -0.213 | -0.213 | -0.213 | -0.213 | -0.213 |
> | lr=0.01 | -0.213 | -0.213 | -0.213 | -0.213 | -0.212 | -0.212 |
> | lr=0.1 | -0.213 | -0.211 | -0.209 | -0.206 | -0.204 | -0.202 |
> | **lr=1** | **-0.213** | **-0.194** | **-0.182** | **-0.175** | **-0.170** | **-0.165** |
> | **lr=10** | **-0.213** | **-0.176** | **-0.168** | **-0.165** | **-0.163** | **-0.162** |
> | lr=100 | diverge risk | | | | | |
> | lr=1000 | diverge risk | | | | | |
>
> **Notably**, leveraging the DataMask framework and engineering efforts such as quality-aware initialization and pruning, a single selection on a split file of 1,000,000 samples requires approximately 0.5–1 hour on a single GPU. **This provides sufficient capacity for searching for the best parameters under a specific objective.**

---

> > ### Author Response · Authors · 2025-11-21
> > **Rebuttal to Reviewer NCCX (2/2)**
> >
> > > W4: The writing in some sections is not clear enough, for example, the explanation of the content in Figure 3.
> >
> > **Reply to W4:**
> >
> > **We thank the constructive comments. Here, we provide the revised explanation for Figure 3:**
> > > Illustration of the quality and diversity scores when optimizing under DATAMASK with two initialization and two optimization strategies. Initialization strategies: same initialization (green and red lines) that all initial sampling probabilities are the same and quality initialization (black and blue lines) that initial sampling probabilities are proportional to the samples' quality score. Optimization strategies: diversity optimization (green and black lines) that only optimizes the diversity metric and joint optimization (blue and red lines) that optimizes both the diversity and quality metrics.
> >
> > **Besides, we provide more details about the two type initialization strategies used in the figure in Appendix A6.2 of the revised submission:**
> > > Initialization Strategies for Logits. As discussed in Figure 3 and Section 4.3 ("Quality-aware Pruning and Initialization"), we investigate two types of initialization strategies for logits: i) Same Initialization and ii) Quality-Aware Initialization. For Same Initialization, all logits are set to 0. For Quality-Aware Initialization, the logit of the i-th sample is initialized as:  $$\text{logit}(x_i)=\frac{f_{\text{qua}}(x_i)-q_{\text{min}}}{q_{\text{max}}-q_{\text{min}}}* (l_{\text{max}}-l_{\text{min}}) + l_{\text{min}}$$, where $f_{\text{qua}}(x_i)$ is the quality score of the sample, and $q_{\text{max}}$, and $q_{\text{min}}$ represent the maximum and minimum quality scores (15 and 0 in our settings). To determine the bounds of the optimized logits, $l_{\text{max}}$ and $l_{\text{min}}$, we referenced the maximum and minimum logit values observed in optimization with Same Initialization setting after convergence. These values were consistently found to be approximately 5 and -5 across most experiments. Through this transformation, the logits are initialized based on their corresponding quality scores. As shown in Figure 3 and Section 4.3, while both strategies yield similar final optimized values, Quality-Aware Initialization significantly reduces the selection time.

---

> > > ### Comment · Reviewer_NCCX · 2025-11-27
> > >
> > > Thanks for the detailed response. I will maintain my score.

---

### Official Review · Reviewer_nve6 · 2025-10-27

**Soundness:** 3
**Presentation:** 2
**Contribution:** 2
**Rating:** 4
**Confidence:** 3

**Summary:**

This paper focus on the large-scale pre-training data selection for LLMs, which aims at optimizing the metrics for like data quality and data diversity. In this paper, the author point out that current open sourced dataset like FineWeb and DCLM rarely consider the data quality and diversity or the more metrics jointly in a single selection process, which leads to these problems based on their research: Selection based on quality metrics will leads to severe diminishing returns during long-turn pre-training, while the selection based on quality will remove to many high-quality samples, which limits the effect of pre-training LLMs. To solve this problem, this paper introduce DATAMASK: a new joint learning framework designed for large-scale pre-training data selection to simultaneously optimize multiple types of metrics (mainly focusing on quality and diversity). By interpreting the selection task as a mask learning problem and using multiple enhance technique, DATAMASK significantly reduces selection time by 98.9% compared with greedy algorithm. And in FineWeb dataset, they select a FineWeb-Mask subset, and achieves significant improvements of 3.2% on a 1.5B dense model and 1.9% on a 7B MoE model after pre-training with hundreds of billions of tokens, demonstrating its effectiveness.

**Strengths:**

Novelty for the problem definition: The paper conceptualizes the large-scale data selection problem into a learnable mask optimization task and use policy gradient-based optimization and various acceleration enhancements to optimize the selection speed.

Strong motivation and empirical analysis: The paper demonstrates the fundamental limitations of single-metric selection on large scale pre-training dataset, and use the visualizations to express the conflict of data quality and diversity that supports the central motivation.

High engineering availability and scalability: Implementation on 15T FineWeb corpus shows the method’s availability in engineering. Decrease 98.5% time cost compare with greedy algorithm, enabling the studies in large-scale datasets shows the scalability more research.

**Weaknesses:**

Limited methodological originality: The novelty is incremental, not a fundamentally new method. The framework of combination of Mask Learning and Policy Gradient is a direct application of Reinforce-style policy gradient to a combinatorial subset selection problem. Similar implementation have been used in: RL-based data pruning or sample selection (e.g., RLDataSampler, ICML 2022);Differentiable subset selection in vision and NLP (e.g., DPPNet, CVPR 2021; SubsetFormer, NeurIPS 2023).

Lack of fair on challenging baselines:All baselines are existing data recipes, but without comparison with recent learning-based selection frameworks like: DataComp-LM (Li et al., NeurIPS 2024) etc. Also missing the comparison with other learnable data selection methods, which weakens the claims of the effectiveness.

Lack of the explainability:Even the paper gives the heat map, the paper didn’t make qualitative analysis and visualization for the selected datas’ types, realms etc.

Weak theoretical justification:The paper only gives the upgrade formula for optimization in the end of method, and then comes into the experiment part, without explanation for convergence, bias introduced by probabilistic relaxation or gradient variance.

**Questions:**

1.What is the difference between DATAMASK and the classical reinforce-based subset selection? Please clarify the difference between traditional reinforce mechanism or active learning.
2.Can the union optimization frame explains to more than two metrics? Like adding text toxic or the other metrics.
3.Can you release more visualizations of selected samples? For example: distribution of domains, text lengths etc.
4.How to confirm the stability for optimization? In the sampling upgrade with larger variance?

---

> ### Comment · Reviewer_nve6 · 2025-11-13
> **Correction of References**
>
> I’m sorry for the oversight — the references I mentioned in my review were incorrect. Please find below the corrected list of references:
>
> 1. RLDataSampler -> DVRL：Data Valuation using Reinforcement Learning.
>
> 2. DPPNet -> DppNet: Approximating Determinantal Point Processes with Deep Networks
>
> 3. SubsetFormer ->  FLEXSUBNET: Neural Estimation of Submodular Functions with Applications to Differentiable Subset Selection
>
> 4. DataComp-LM -> Data Selection for Language Models via Importance Resampling
>
> Apologies again for the confusion caused by my earlier mistake, and thank you for bringing this to my attention.

---

> > ### Author Response · Authors · 2025-11-13
> > **Thank for the corrections and look forward to discussing later!**
> >
> > We thank the reviewer for the time spent reviewing our work and for providing constructive comments. With corrected references, we look forward to further discussion during the subsequent discussion phase.
> >
> > Thank again,
> > The Authors of Submission 10572.

---

> ### Author Response · Authors · 2025-11-21
> **Rebuttal to Reviewer nve6 (1/5)**
>
> Rebuttal to Reviewer nve6
>
> **Thanks for the constructive feedback provided by the Reviewer nve6. We sincerely appreciate the time and effort you dedicated to evaluating our work. Below, we provide detailed responses to the weaknesses and questions, which we denote as W and Q. Besides, we also added the following discussions into revised submission.**
>
> > **W1 and Q1:** Limited methodological originality: The novelty is incremental, not a fundamentally new method. The framework of combination of Mask Learning and Policy Gradient is a direct application of Reinforce-style policy gradient to a combinatorial subset selection problem. Similar implementation have been used in: RL-based data pruning or sample selection (e.g., RLDataSampler, ICML 2022);Differentiable subset selection in vision and NLP (e.g., DPPNet, CVPR 2021; SubsetFormer, NeurIPS 2023). 1.What is the difference between DATAMASK and the classical reinforce-based subset selection? Please clarify the difference between traditional reinforce mechanism or active learning. Revised ref:
> RLDataSampler -> DVRL：Data Valuation using Reinforcement Learning.
> DPPNet -> DppNet: Approximating Determinantal Point Processes with Deep Networks
> SubsetFormer -> FLEXSUBNET: Neural Estimation of Submodular Functions with Applications to Differentiable Subset Selection
>
> **Reply to W1 and Q1:**
> 1. **Differences to the corrected references.** In the following, we provide a comparison against the corrected reference papers across several metrics: **scale, methodology, and computation.**
> - **Scale.** As shown in the following table, all reference methods operate on a small scale. Our data scale is up to 150,000 times larger, and our model scale is 7,000 times larger than the scope of the reference methods.
> - **Methodology.** These reference methods either require a small validation set (DVRL), utilize an additional trained model (DPPNet), or impose functional requirements (FLEXSUBNET). These methods typically use a trained model to produce sampling probabilities for a given batch of data. In sharp contrast, our method directly optimizes the mask probabilities for all samples based on any specified function.
> - **Computation.** Notably, compared to DVRL, which required 8 hours to select only 0.06M samples, our selection of 1M samples only requires 30 minutes. This difference makes our approach far more feasible in the current large data and large model paradigm.
>
> Considering methodology, these methods typically use a trained model to produce sampling probabilities for a given batch of data. In sharp contrast, our method directly optimizes the mask probabilities for all samples based on any specified function. **We perform a direct comparison of our DATAMASK to DVRL (the most cited paper) using Pair-wise Similarity to analyze their advantages and shortcomings in the real application of large-scale pre-training data. Details see the response in the next question.**
>
> | | Data | Data scale | Trained Model | Model scale | Method | computation |
> | :--- | :--- | :--- | :--- | :--- | :--- | :--- |
> | DVRL | Cifar100 | 0.06M | ResNet-32 | <1 M | Training the data value estimator using a reinforcement signal of the reward obtained on a small validation set that reflects performance on the target task. | 8 hour on 0.06M samples |
> | DPPNet | MNIST, Celeb | 0.06M | Wide-ResNet | <1 M | Develop an attention mechanism based on the transformer that captures a notion of dissimilarity between feature vectors | - |
> | FLEXSUBNET | Amazon baby registry dataset | 0.1M | neural network | <1 M | Flexible neural models for both monotone and non-monotone submodular functions for selection. | - |
> | Ours | FineWeb | 15,000M | 1/7 B LLM | 1-7,000 M (1-7B) | Mask learning with any function without the need of validation set or additional model | 40 minutes on 1M samples |
>
>
> 2. Relationship and difference with reinforce mechanism or active learning.
>
> Policy gradient estimation (PGE) is used in reinforcement learning to solve Expected Cumulative Reward, and is also used in our mask learning framework to solve large scale data selection with diversity score in a set function formulation. Different to reinforce mechanisms, we do not have environment, trajectory, action, states, and critic or value models. Our DATAMASK is an optimization framework with PGE that jointly optimizes multiple criteria more effectively and replaces active learning (based on human-designed heuristics) with a learned policy.

---

> ### Author Response · Authors · 2025-11-21
> **Rebuttal to Reviewer nve6 (2/5)**
>
> > **W2:** Lack of fair on challenging baselines:All baselines are existing data recipes, but without comparison with recent learning-based selection frameworks like: DataComp-LM (Li et al., NeurIPS 2024) etc. Also missing the comparison with other learnable data selection methods, which weakens the claims of the effectiveness.
>
> **Reply to W2:**
>
> 1. **In the original submission,the learning-based selection framework DataComp-LM is introduced as DCLM and has already been compared in our experiments denoted as FineWeb-DCLM.** The results are provided in Table 2 of the submission and are partially shown below:
>
> | 1.5 B | **Reading Comprehension** | **Reasoning** | **World Knowledge** | Avg |
> | :--- | :--- | :--- | :--- | :--- |
> | FineWeb | 46.0 | 53.6 | 38.8 | 44.9 |
> | FineWeb-DCLM | 48.2 | **57.2** | 39.5 | 46.9 |
> | FineWeb-Mask(ours) | **48.8** | 56.6 | **42.1** | **48.1** |
> | **7B MoE** | **Reading Comprehension** | **Reasoning** | **World Knowledge** | Avg |
> | FineWeb | 48.1 | 58.1 | 46.8 | 50.7 |
> | FineWeb-DCLM | 49.0 | **60.4** | 47.8 | 52.2 |
> | FineWeb-Mask(ours) | **49.1** | 59.9 | **48.9** | **52.6** |
>
>
>
> 2. **Compared to leanable method DVRL.** To facilitate this comparison, we follow the DVRL pipeline to train a Data Value Estimator, which consists of three linear modules with ReLU activation, followed by a final sigmoid activation. It was trained on 1,000,000 text samples, each with 768 dimensions of features, using a reward based on Pair-wise Similarity (averaged cosine similarity). After convergence, the Data Value Estimator predicted the logits for these 1,000,000 samples as well as for another 1,000,000 unseen samples. Based on these predictions, we selected the top 10% of samples and recorded their exact pairwise similarity compared with our DATAMASK framework. In the following table, we present the pairwise similarity value **(lower the better)** of selected data and the time taken to train the model until full convergence.
>
> | | **Before optimization** | **Seen Set<br>(after optimization)** | **Unseen Set<br>(after optimization)** | **Computational time** |
> | :--- | :--- | :--- | :--- | :--- |
> | DVRL | 0.7101 | 0.6874 | 0.7040 | 2h30min |
> | DATAMASK | 0.7101 | 0.6831 | - | 40min |
>
>
> Compared to our approach to training all samples with unique probabilities, methods like DVRL train a small model to predict the probabilities for a batch of samples. However, during our implementation of DVRL with large-scale pre-training text data, they show the following issues:
> - **Sub-optimal performance on the seen set.** Unlike our DATAMASK framework, which directly optimizes the sampling probabilities of all samples based on the metric, DVRL trains a model to predict the probabilities for a given batch of data. This approach amplifies the errors arising from the model's learning capacity, achieving much worse converged values after optimization.
> - **Poor generalization ability on the unseen set and huge computation.** We observe significantly larger scores on the unseen set compared to the seen set, revealing a lack of generalization in DVRL. This implies that despite DVRL training a model to predict the probabilities, the model must be re-trained when handling new data. Given that a single DVRL model already requires significantly more computation (6x) than our DATAMASK, the need for repeated retraining renders DVRL computationally infeasible for selecting large-scale pre-training data.
> - **Unstable training.** Following DVRL's pipeline, we identified significant divergence risks at the 1M sample scale. After careful selection of hyperparameters and engineering efforts , we finally achieved relatively satisfactory optimized values.
>
> **The comparison is added in Appendix A.8 of revised submission, and we hope these address the reviewer's concern.**

---

> > ### Author Response · Authors · 2025-11-21
> > **Rebuttal to Reviewer nve6 (3/5)**
> >
> > > **W3 and Q3:** Lack of the explainability:Even the paper gives the heat map, the paper didn’t make qualitative analysis and visualization for the selected datas’ types, realms etc. 3.Can you release more visualizations of selected samples? For example: distribution of domains, text lengths etc.
> >
> > **Reply to W3 and Q3:**
> >
> > We thank the reviewers for their constructive comments regarding the need for more qualitative analysis and visualization to improve our submission. **As requested, the detailed distributions of domains and text lengths for the selected data are provided in the following. More formal descriptions and clearer figures have been included in Appendix A6 of revised submission.**
> >
> > 1. **Detailed distributions of text lengths.** Here we provide the text length distribution of FineWeb, FineWeb-Edu, and our FineWeb-Mask. As shown in following table and Figure 15 of revised submission, **the token length of the selected FineWeb-Mask documents falls between that of the raw FineWeb and FineWeb-Edu.** This observation demonstrates a successful trade-off between quality (FineWeb-Edu) and diversity (raw FineWeb distribution).
> >
> > | Token length(TL) | 0<TL<200 | 200<TL<400 | 400<TL<600 | 600<TL<800 | 800<TL<1000 | 1000<TL<2000 |
> > | --- | --- | --- | --- | --- | --- | --- |
> > | Fineweb | 24.69% | 25.58% | 15.92% | 10.46% | 6.67% | 11.77% |
> > | Fineweb-mask | 12.9% | 23.0% | 17.06% | 12.69% | 8.89% | 17.29% |
> > | Fineweb-edu | 8.51% | 21.14% | 17.27% | 14.22% | 9.91% | 19.49% |
> > | | **2000<TL<4000** | **4000<TL<6000** | **6000<TL<8000** | **8000<TL<10000** | **>10000** |
> > | Fineweb | 3.53% | 0.74% | 0.29% | 0.14% | 0.21% |
> > | Fineweb-mask | 5.62% | 1.28% | 0.54% | 0.29% | 0.44% |
> > | Fineweb-edu | 6.57% | 1.48% | 0.61% | 0.32% | 0.48% |
> >
> > 2. **Detailed distributions of text domains.** Since the original FineWeb dataset only provides the year of the CC snapshot to which each sample belongs, but not the text domains, we randomly sampled 1,000,000 samples from FineWeb, FineWeb-Edu, and our FineWeb-Mask, respectively. We then used a recently released model TopicClassifier (https://huggingface.co/WebOrganizer/TopicClassifier) to rate their domains. The detailed distributions are provided in the following table. The categories are ordered by the sample number of FineWeb-Edu. We also provide a more formal figure in the revised submission (Figure 16). **Results show that FineWeb-Edu is skewed towards Science & Tech., Health, and History domains. Conversely, our approach demonstrates the successful trade-off between quality (FineWeb-Edu) and diversity (raw FineWeb distribution).**
> >
> > | | Science & Tech. | Health | Education & Jobs | History | Politics | Literature | Religion | Home & Hobbies |
> > | :--- | :--- | :--- | :--- | :--- | :--- | :--- | :--- | :--- |
> > | **Fineweb** | 3.48% | 6.42% | 6.11% | 1.60% | 5.79% | 2.93% | 3.11% | 7.18% |
> > | **Fineweb-edu** | 23.32% | 19.94% | 10.54% | 9.54% | 4.66% | 3.91% | 3.73% | 3.35% |
> > | **Fineweb-mask** | 12.85% | 9.08% | 6.22% | 7.21% | 11.28% | 4.77% | 3.71% | 2.22% |
> > | | Industrial | Finance & Business' | Art & Design | Software Dev. | Software | Transportation | Crime & Law | Food & Dining |
> > | **Fineweb** | 2.57% | 8.41% | 3.25% | 1.88% | 3.63% | 3.37% | 3.55% | 3.86% |
> > | **Fineweb-edu** | 3.14% | 2.31% | 2.22% | 2.03% | 1.67% | 1.63% | 1.42% | 1.36% |
> > | **Fineweb-mask** | 1.73% | 4.14% | 3.02% | 2.60% | 2.52% | 2.66% | 3.63% | 1.86% |
> > | | Sports & Fitness | Social Life | Hardware | Entertainment | Travel | Games | Fashion & Beauty | Adult |
> > | **Fineweb** | 7.57% | 4.41% | 2.39% | 8.07% | 3.25% | 2.66% | 3.65% | 0.87% |
> > | **Fineweb-edu** | 1.23% | 0.90% | 0.89% | 0.83% | 0.69% | 0.44% | 0.25% | 0.01% |
> > | **Fineweb-mask** | 4.26% | 1.41% | 1.38% | 9.34% | 1.58% | 1.97% | 0.47% | 0.10% |
> >
> >
> > 3. **Notably, some visualizations were also provided in the original submission,** which may also be helpful for understanding the statistics:
> > - t-SNE visualization of selected sample features in Figure 2.
> > - Quality and diversity optimization in Figure 3.
> > - Heatmap of quality score and diversity score in 400 clusters in Figure 5.
> > - Average token length of selected samples compared to the raw dataset in Figure 8.

---

> ### Author Response · Authors · 2025-11-21
> **Rebuttal to Reviewer nve6 (4/5)**
>
> > W4 and Q4: Weak theoretical justification:The paper only gives the upgrade formula for optimization in the end of method, and then comes into the experiment part, without explanation for convergence, bias introduced by probabilistic relaxation or gradient variance. 4.How to confirm the stability for optimization? In the sampling upgrade with larger variance?
>
> **Reply to W4 and Q4:**
>
> For convergence and to confirm the stability for optimization, we made following two efforts from both empirical and theoretical views:
>
> 1. **Empirical view.** We additionally provided the ablation study with error analysis during optimization in the Appendix A9 to better demonstrate convergence behavior and stability. As illustrated in the Figure 12 from the revised submission, the variance of the optimization process across different subsets is small, and for different seeds on the same subset, the variance is negligible.
> 2. **Theoretical view.** Our proposed gradient formulation with group relative reward achieves a lower variance than the original policy gradient estimation, which in turn accelerates training and achieves better optimization stability. By choosing a sufficiently large group number G (as verified in Figure 7 and Section 4.3), we can ensure the stability for optimization in the sampling upgrade with larger variance.  Following is the detailed analysis:
>
> We first consider the expectation form of Equation (10), which constructs the average gradient under normalized loss with independently sampled masks. For any component $j$, we have:
> $$\mathbb{E}_{P(M|L)}[\mu_G\nabla\log(P(M|L))]=\mu_G\int P(M|L)\frac{\nabla P(M|L)}{P(M|L)}dM=\mu_G\nabla\int P(M|L)dM=\mu_G\nabla 1=0.$$
>
> Therefore, the expectation of its mean is always zero. By denoting the sample loss with respect to $M$ as a function $f(M)$, we have:
> $$\mathbb{E}[\frac{1}{G}\sum_{j=1}^{G}(\frac{f(M_j)-\mu_G}{\sigma_G}\nabla\log(P(M_j|L)))]=\frac{1}{\sigma_G}\frac{1}{G}\sum_{j=1}^{G}\mathbb{E}_{P(M|L)}f(M_j)\nabla\log(P(M_j|L)).$$
>
> The proposed policy gradient is essentially the mean of the original policy gradients over all samples within the group (differ by a constant factor $\sigma_G$). Therefore, its gradient direction is an unbiased estimate.
>
> Next, we consider the variance. Using the variance expansion formula, we can directly write:
> $$\text{Var} = \mathbb{E}[\frac{1}{G}\sum_{j=1}^{G}(\frac{f(M_j)-\mu_G}{\sigma_G}\nabla\log(P(M_j|L)))]^2 - [\mathbb{E}[\frac{1}{G}\sum_{j=1}^{G}(\frac{f(M_j)-\mu_G}{\sigma_G}\nabla\log(P(M_j|L)))]]^2.$$
> All cross terms vanish because the different masks are sampled independently, and therefore their covariance terms are zero. Then we consider the variance difference between our proposed PGE and vanilla PGE:
> $$\Delta\text{Var} = \mathbb{E}\frac{1}{\sigma_G^2}\frac{1}{G^2}\sum_{j=1}^{G}\nabla\log^2(P(M_j|L))[(f(M_j)-\mu_G)^2]-f(M_j)^2].$$
> It is clear that $\nabla\log^2(P(M|L))$ must be greater than zero. The remaining terms can be further simplified to $$(f(M_j)-\mu_G)^2]-f(M_j)^2=-\mu_G(2f(M_j)-\mu_G)$$ where $\mu_G$ is the average of the values $f(M_j)$ in group G. A strict analysis of this formula is challenging because the distribution of $f(M)$ may differ from the distribution of $M$ itself. However, by taking the Gaussian case as an example, we can illustrate the following conclusion. Assume that $f(M)$ follows a normal distribution $\mathcal{N}(\mu_G, \sigma_G)$, then we have:
> $$P(\frac{1}{G}\sum_i (2f(M_j)-\mu_G)>0)=\Phi(\frac{\mu_G\sqrt{G}}{2\sigma_G}),$$
> where $\Phi$ is the CDF of the Gaussian distribution.
>
> **Overall, our proposed policy gradient can be regarded as a novel update formulation that is directionally unbiased and achieves variance reduction $\Delta\text{Var}<0$ with high probability $\Phi(\frac{\mu_G\sqrt{G}}{2\sigma_G})$. This property can substantially accelerate the training process and ensure the optimization stability, especially in large-scale combinatorial optimization or distribution-learning tasks.**

---

> > ### Author Response · Authors · 2025-11-21
> > **Rebuttal to Reviewer nve6 (5/5)**
> >
> > > **Q2:** 2.Can the union optimization frame explains to more than two metrics? Like adding text toxic or the other metrics.
> >
> > **Reply to Q2:**
> >
> > **Our framework can union any types of metric, which can be formulated as follows:**
> >
> > $\sum_{i}\lambda_i f_i(U), s.t. \sum_{i}\lambda_i=1$,
> >
> > where $f_i(U)$ can be any score type defined on a given data subset $U$. For example, **considering the requested metric of text toxicity,** we prompt a large LM to collect samples with safety scores. Safe documents are rated at $1$, while documents that may contain adult, hateful, or privacy-related content are rated with $0$. Then, we use them to train a small 0.6B LM (such a rator is not specifically trained during this rebuttal), and apply this smaller LM to rate all documents. Using this text toxicity metric, we design the union optimization objective as:
> >
> > $$0.4 * f_\text{qua} (U)+0.4 * f_\text{div} (U)+0.2 * f_\text{toxic} (U)$$
> >
> > where $f_\text{qua}, f_\text{div}, f_\text{toxic}$ are respectively quality metric, diversity metric, and text toxic metric. The optimized values obtained during mask learning are presented in the following table. It can be seen that **all metrics increase with continued iterations, which demonstrates the effectiveness of our method. Notably, the initial value of toxic metric equals to 0.99 means that 99% of the selected samples contain safe content. After optimization, this percentage increases to 99.96\%**. Users can simply adjust the balancing ratio to control the trade-off among all metrics based on their specific optimization target for the given data.
> >
> > | Higher the better | Round = 0 | 100 | 500 | 1000 | 5000 |
> > | :--- | :--- | :--- | :--- | :--- | :--- |
> > | Quality metric | 0.180 | 0.188 | 0.230 | 0.277 | 0.400 |
> > | Diversity metric | -0.300 | -0.301 | -0.294 | -0.292 | -0.290 |
> > | Toxic metric<br>(safety score) | 0.9900 | 0.9961 | 0.9995 | 0.9996 | 0.9996 |

---

> ### Comment · Reviewer_nve6 · 2025-11-26
>
> Thanks for the reply, I will raise my score to 6.

---

> > ### Author Response · Authors · 2025-11-26
> > **Thanks for the reply and the increased score !**
> >
> > We appreciate the increased score from Reviewer nve6 and will continue to improve our submission according to your advice.

---

### Official Review · Reviewer_iTj3 · 2025-10-31

**Soundness:** 3
**Presentation:** 3
**Contribution:** 2
**Rating:** 6
**Confidence:** 4

**Summary:**

The paper identifies that existing pre-training data selection methods optimize quality or diversity separately, leading to either semantic redundancy or loss of valuable samples, which limits LLM performance. To address this, the authors introduce DATAMASK, a novel framework that jointly optimizes multiple metrics. Applied to FineWeb, DATAMASK produces FineWeb-Mask (1.5 trillion tokens), which demonstrates 3.2% and 1.9% performance improvements on 1.5B and 7B (MoE) models respectively. The work proves that balanced quality-diversity optimization is both computationally feasible and empirically superior for large-scale pre-training data selection.

**Strengths:**

- There is an inherent trade-off between generality and specificity that has not been considered in existing related work.
- I appreciate the fomalized approach that provides users with a more principled way of data curation. I believe such techniques are particularly valuable in increasing the sample efficency during pre-training and ultimately driving down cost.
- The transparent cost breakdown helps others estimate whether datamask is a useful (and affordable) technique for their individual use cases.

**Weaknesses:**

- When arguing about pre-training the proposed dataset the FineWeb-Mask rather small for fully training 7/8B parameter (dense) or even larger models. SOTA 8b dense models are typically trained on 10T+ tokens. I could see the dataset to be applicable for what sometimes is refered to stage-two pre-training, i.e., showing documents to a model that contain desirable information for later post-training steps that require versatility. Exploring how well the specificity-/generality-balance introduced through datamask would make a useful addition to the paper.
- The paper misses a recent work on a pre-trainign dataset that is also derived from fineweb. Even though, it's based on heuristics, the data processing pipeline makes an effort to carefully balance specificity and generality [1].

**Sources:**

[1] GneissWeb: Preparing High Quality Data for LLMs at Scale, Gohari et al., 2025

**Questions:**

- What does "optimized score" refer to in the G ablation study? I understand that G is used to keep the computational costs in check but doesn't it also implicitly change the optimization objective later on because of the dependency among samples in a group?
- Out of curiosity, is there any notable performance benefits for post-training when using datamask over vanilla fineweb?
- Will you open-source the data curation recipe (code) if the paper gets accepted?

---

> ### Author Response · Authors · 2025-11-21
> **Rebuttal to Reviewer iTj3 (1/2)**
>
> **Thanks for the constructive feedback provided by the Reviewer iTj3. We sincerely appreciate the time and effort you dedicated to evaluating our work. Below, we provide detailed responses to the weaknesses and questions, which we denote as W and Q. Besides, we also added the following discussions into revised submission.**
>
> > **W1 and Q2:** When arguing about pre-training the proposed dataset the FineWeb-Mask rather small for fully training 7/8B parameter (dense) or even larger models. SOTA 8b dense models are typically trained on 10T+ tokens. I could see the dataset to be applicable for what sometimes is refered to stage-two pre-training, i.e., showing documents to a model that contain desirable information for later post-training steps that require versatility. Exploring how well the specificity-/generality-balance introduced through datamask would make a useful addition to the paper. Out of curiosity, is there any notable performance benefits for post-training when using datamask over vanilla fineweb?
>
> **Reply to W1:**
>
> 1. **Stage 2 pretraining following post training results.** As requested, we tested our FineWeb-Mask dataset on a stage-two pre-training process, followed by post-training steps. The results are presented in the following table. The base model was initially pre-trained on 400B tokens of FineWeb-Raw data. We then performed the stage-two pre-training using 200B tokens of FineWeb-Mask and FineWeb-Edu. Then, the subsequent post-training was conducted on recently open sourced dataset: tulu-3-sft-mixture (https://huggingface.co/datasets/allenai/tulu-3-sft-mixture). The results demonstrate the **promising performance** of our method and the selected data in **stage-two pre-training followed with post training.** Furthermore, the balance between the educational score and the diversity metric **also benefits subsequent post-training. We thank the reviewer for their constructive comments, which provided a useful addition to the submission. These results have been included in Appendix A.7 of revised submission.**
>
> | 7B MoE | Race-H | Race-M | HellaSwag | NQ | OBQA | KQAPro | MMLU pro | TrivalQA | ARC-C | SIQA | PIQA | WinoGrande | avg | win |
> | :--- | :--- | :--- | :--- | :--- | :--- | :--- | :--- | :--- | :--- | :--- | :--- | :--- | :--- | :--- |
> | FineWeb-Edu (after SFT) | 50.3 | **64.5** | **74.6** | 21.1 | 58.6 | 51.2 | 47.5 | 65.5 | **55.8** | 57.6 | 81.2 | 70.2 | 59.1 | 3/12 |
> | FineWeb-Mask (after SFT) | **50.7** | 64.3 | **74.6** | **22.4** | 57 | **54.7** | **47.8** | **69.6** | 54.8 | **58.4** | **81.8** | **70.6** | **60.2** | **9/12** |
>
> 2. **The size explanation of FineWeb-Mask (about 1.5T tokens).**
> - **In the context of scaling laws**, pre-training a compute-optimal 7–8B LLM requires 140–160 billion tokens. Our FineWeb-Mask contains 1.5 trillion tokens, a volume sufficient to **support the training of models up to 70B in size from the view of scaling law.** Notably, all baselines, including the recommended related work GneissWeb, pre-train their 7B models on approximately 300–400 billion tokens.
> - **For a fair comparison, FineWeb-Edu, FineWeb-Pro, and DCLM-Baseline all operate within the 100 billion to 2 trillion token scale in the initial pre-training stage.** As shown in Figure 11, after being pre-trained on 300 billion tokens, most results among different methods have either nearly converged or separated clearly, which is sufficient for comparing the methods effectively.
> - **To the wider research community,** pre-training a 7B model on over 15T tokens is impractical for sharing academic methods and reproducation . Such an endeavor requires more than 500,000 H100 hours (approximately 150 days on 128 H100 GPUs).
>
> **And notably, our method is scalable to select  samples at 15T token scales.**

---

> > ### Author Response · Authors · 2025-11-21
> > **Rebuttal to Reviewer iTj3 (2/2)**
> >
> > > W2: The paper misses a recent work on a pre-training dataset that is also derived from fineweb. Even though, it's based on heuristics, the data processing pipeline makes an effort to carefully balance specificity and generality [1].(GneissWeb: Preparing High Quality Data for LLMs at Scale)
> >
> > **Reply to W2:**
> >
> > **We thank the reviewers for their constructive comments regarding this related work.** GneissWeb has been added to our revised submission as a related work. GneissWeb utilizes substring deduplication **followed** by an ensemble quality filter. In contrast, we apply document selection considering **both** semantic similarity and quality filter scores. Substring deduplication is orthogonal to our work (especially the semantic selection part); furthermore, it changes the content of the original document and cannot handle semantic duplication across documents.
> >
> > Notably, we were unable to find the curated data on their website and dataset card from Hugging Face, thus hindering a direct comparison with this pipeline. **Therefore, in the following table, we make a primary comparison** of exact deduplication strategy followed by quality scores used in the submission on FineWeb dataset. Results show sub-optimal performance of this strategy compared to our method.
> >
> > | 7B MoE 300B tokens | Race-H | Race-M | HellaSwag | NQ | OBQA | KQAPro | MMLU pro | TrivalQA | ARC-C | SIQA | PIQA | WinoGrande | avg | win |
> > | :--- | :--- | :--- | :--- | :--- | :--- | :--- | :--- | :--- | :--- | :--- | :--- | :--- | :--- | :--- |
> > | Fineweb | 42.1 | 54.1 | **69.9** | 17.9 | 53.6 | 49.8 | 35.8 | 53.8 | 40.7 | **52.8** | **78.6** | 63.7 | 50.7 | 3/12 |
> > | Exact dedup + quality filters | 42.2 | 55.0 | 67.7 | **21.9** | 53.4 | **51.8** | 38.8 | 58.6 | 42.3 | 50.9 | 78.0 | 65.0 | 52.1 | 2/12 |
> > | Ours | **42.8** | **55.4** | 66.1 | 19.5 | **55.0** | 51.4 | **39.5** | **61.7** | **45.9** | 51.2 | 77.5 | **65.1** | **52.6** | **7/12** |
> >
> >
> > > Q1: What does "optimized score" refer to in the G ablation study? I understand that G is used to keep the computational costs in check but doesn't it also implicitly change the optimization objective later on because of the dependency among samples in a group?
> >
> > **Reply to Q1:**
> >
> > 1. The term "Optimized Score" here refers to the value of the diversity score (facility location) measured during the optimization process. **We have made it more clear in the revised submission.**
> > 2. **G is not used to reduce or keep the computational cost. Instead, it is used to help estimate the gradient (similar to the group relative reward used in GRPO[1]) and is not involved in calculating the optimization objective.** As shown in Figure 7 of the submission, a smaller G leads to unstable optimization, indicating that the gradient estimation is inaccurate. However, a larger G inherently incurs a higher computational cost, and we also demonstrate the appropriate selection for G in Figure 7.
> >
> > [1]DeepSeekMath: Pushing the Limits of Mathematical Reasoning in Open Language Models
> >
> > > Q3: Will you open-source the data curation recipe (code) if the paper gets accepted?
> >
> > **Reply to Q3:**
> >
> > **Yes, we promise to release the code for the data curation recipe as well as the FineWeb-Mask dataset.**

---

> > > ### Comment · Reviewer_iTj3 · 2025-11-24
> > >
> > > Thank you for the thorough rebuttal and answering my questions. I especially appreciate the additional experiments and comparisons given the large amount of compute needed to do that in such a short amount of time. I have updated my rating.

---

> > > > ### Author Response · Authors · 2025-11-24
> > > > **Thanks for the reply and the increased score !**
> > > >
> > > > We are grateful for the endorsement of the Reviewer iTj3 and the increased score! We will keep improving our submission, following your valuable suggestions.

---

### Official Review · Reviewer_5FG7 · 2025-11-01

**Soundness:** 3
**Presentation:** 2
**Contribution:** 3
**Rating:** 6
**Confidence:** 3

**Summary:**

This paper purpose DATAMASK, a new framework for large-scale data selection method based on policy-gradient mask learning framework. It targets jointly optimizes the quality and diversity in trillion-token constraints. The authors use differentiable optimization instead of probabilistic masks, and run experiments on FineWeb datasets, achieving a 98.9% reduction in selection cost while not sacrificing the performance. Evaluated across 12 diverse downstream tasks, this subset achieves significant performance gains of 3.2% on a 1.5B dense model and 1.9% on a 7B MoE model, demonstrating its effectiveness.

**Strengths:**

Originality: The paper addresses a critical problem, namely the trade-off between quality and diversity in LLM pre-training data selection. The idea of treating data selection as a mask learning problem and using policy gradients for optimization is novel. It moves beyond traditional greedy selection strategies, offering a unified learning-based approach.
Quality: The paper is supported by rigorous empirical validation, including large-scale experiments on trillion-token datasets. Ablation studies are provided to analyze the impact of different diversity metrics and hyperparameters (e.g., λ, G). The method is evaluated on 12 diverse downstream tasks, demonstrating consistent and significant improvements.
Clarity: The paper is well-structured and clearly written. The core mechanism is illustrated with formulas and algorithmic descriptions. The use of visualizations (e.g., t-SNE plots, heatmaps) helps show key insights effectively.
Significance: The proposed method has practical value for improving LLM pre-training efficiency and performance. It opens up a new direction for joint optimization of multiple data metrics, which could influence future data curation paradigms. The released FineWeb-Mask dataset is a valuable resource for the community.

**Weaknesses:**

1.	Partial Ablation of Core Parameters
While the paper ablates diversity metrics, the balancing hyperparameter (λ), and the group size (G) in policy gradient estimation, it lacks systematic exploration of other key hyperparameters, such as the learning rate, the number of update epochs, and the initialization strategies for logits. This omission limits the understanding of the method's robustness and sensitivity to its full configuration.
2.	Insufficient Accessibility and Clarity in Figures
Figure 2 (t-SNE) uses colors that are not colorblind-friendly, and the legend is small with ambiguous labels. Figure 3 (score evolution) lacks units on the x-axis and error bars/confidence intervals, making it difficult to interpret convergence behavior and stability.
3.	Missing Comparison with Scalable Learned Selection Paradigms
Comparisons are made primarily against heuristic baselines. A comparison with other learned selection methods that are scalable to trillion-token datasets (if available) is needed to better situate DATAMASK's advantages. For methods limited to small-scale data, this could be framed as future work instead of a current limitation.

**Questions:**

1.Beyond computational constraints, the key hyperparameters like the learning rate and number of update epochs are excluded from ablation. Was wondering if it is due to observed insensitivity in preliminary experiments. It will be great to analyze their expected impact on DATAMASK’s performance and convergence?
2. It seems valuable to compare DATAMASK with learned data selection strategies that are scalable to trillion-token datasets. For methods limited to small-scale data, could you elaborate on why they are not suitable for direct comparison, or outline how DATAMASK might outperform them at scale?

---

> ### Author Response · Authors · 2025-11-21
> **Rebuttal to Reviewer 5FG7 (1/3)**
>
> **Thanks for the constructive feedback provided by the Reviewer 5FG7. We sincerely appreciate the time and effort you dedicated to evaluating our work. Below, we provide detailed responses to the weaknesses and questions, which we denote as W and Q. Besides, we also added the following discussions into revised submission.**
>
>
> > **W1 and Q1:**  Partial Ablation of Core Parameters While the paper ablates diversity metrics, the balancing hyperparameter (λ), and the group size (G) in policy gradient estimation, it lacks systematic exploration of other key hyperparameters, such as the learning rate, the number of update epochs, and the initialization strategies for logits. This omission limits the understanding of the method's robustness and sensitivity to its full configuration. Beyond computational constraints, the key hyperparameters like the learning rate and number of update epochs are excluded from ablation. Was wondering if it is due to observed insensitivity in preliminary experiments. It will be great to analyze their expected impact on DATAMASK’s performance and convergence?
>
> **Reply to W1 and Q1:**
>
> We thank the reviewers for their constructive comments regarding the need for detailed information on logit initialization strategies, learning rates, and update epochs. While these hyper-parameters significantly influence performance under a limited selection time budget, the final results are relatively insensitive to them when the time budget is unlimited. **We provide the requested details below, and have incorporated a full discussion into Appendix A6 of the revised submission.**
>
> 1. **Initialization Strategies for Logits.** As discussed in Figure 3 and Section 4.3 ("Quality-aware Pruning and Initialization"), we investigate two types of initialization strategies for logits: i) Same Initialization and ii) Quality-Aware Initialization. For Same Initialization, all logits are set to 0. For Quality-Aware Initialization, the logit of the i-th sample is initialized as:  $$\text{logit}(x_i)=\frac{f_{\text{qua}}(x_i)-q_{\text{min}}}{q_{\text{max}}-q_{\text{min}}}* (l_{\text{max}}-l_{\text{min}}) + l_{\text{min}}$$, where $f_{\text{qua}}(x_i)$ is the quality score of the sample, and $q_{\text{max}}$, and $q_{\text{min}}$ represent the maximum and minimum quality scores (15 and 0 in our settings). To determine the bounds of the optimized logits, $l_{\text{max}}$ and $l_{\text{min}}$, we referenced the maximum and minimum logit values observed in optimization with Same Initialization setting after convergence. These values were consistently found to be approximately 5 and -5 across most experiments. Through this transformation, the logits are initialized based on their corresponding quality scores. As shown in Figure 3 and Section 4.3, while both strategies yield similar final optimized values, Quality-Aware Initialization significantly reduces the selection time.
>
> 2. **Learning rate.** For the learning rate used during joint selection, we conducted a grid search ranging from 0.001 to 1000. The following table presents the results based on the DiSF metric (a measure of diversity). We also provide the optimization curves in Figure 14 of the Appendix in the revised submission. From these results, we observed that an excessively large learning rate (e.g., >100) led to divergence, while an overly small learning rate (e.g., <0.1) significantly hindered convergence speed. Notably, a wide range of learning rates (from 1 to 10) successfully achieves satisfactory results (outperforming the greedy algorithm) while maintaining rapid convergence. Consequently, we adopted a learning rate of 10 for our experiments.
>
> | | 0 step | 2000 step | 4000 step | 6000 step | 8000 step | 10000 step |
> | :--- | :--- | :--- | :--- | :--- | :--- | :--- |
> | lr=0.001 | -0.213 | -0.213 | -0.213 | -0.213 | -0.213 | -0.213 |
> | lr=0.01 | -0.213 | -0.213 | -0.213 | -0.213 | -0.212 | -0.212 |
> | lr=0.1 | -0.213 | -0.211 | -0.219 | -0.206 | -0.204 | -0.202 |
> | **lr=1** | **-0.213** | **-0.194** | **-0.182** | **-0.175** | **-0.170** | **-0.165** |
> | **lr=10** | **-0.213** | **-0.176** | **-0.168** | **-0.165** | **-0.163** | **-0.162** |
> | lr=100 | diverge risk | | | | | |
> | lr=1000 | diverge risk | | | | | |
>
> **See next page for more response to this question**

---

> > ### Author Response · Authors · 2025-11-21
> > **Rebuttal to Reviewer 5FG7 (2/3)**
> >
> > 3. **Update Epochs.** As illustrated in Figure 3 of the submission, selecting 10% of the 1 million samples in our setting (FineWeb-Mask) requires approximately 7,000 to 10,000 update steps. This process takes between 30 minutes and 1 hour on a single GPU, representing a reduction in computational cost of at least 98.9% compared to the classical greedy algorithm. The exact number of update epochs in other settings depends heavily on the user's target optimization values and hyper-parameter configurations. Consequently, most of our ablation studies in Section 4.3—including those on group number, splitting size, pruning, and initialization—aim to minimize the number of epochs required to rapidly reach satisfactory optimized values.
> >
> > Notably, by leveraging the DataMask framework and engineering efforts such as quality-aware initialization and pruning, a single selection run on 1,000,000 samples requires approximately 0.5–1 hour on a single GPU. **This efficiency provides ample opportunity to search for the optimal parameters for specific objectives.**
> >
> >
> >
> > > W2: Insufficient Accessibility and Clarity in Figures. Figure 2 (t-SNE) uses colors that are not colorblind-friendly, and the legend is small with ambiguous labels. Figure 3 (score evolution) lacks units on the x-axis and error bars/confidence intervals, making it difficult to interpret convergence behavior and stability.
> >
> > **Reply to W2:**
> >
> > **Thank the reviewer for the constructive comments, and we have improved these in the revised submission. Please check the following details and let us know if there are any remaining concerns regarding their accessibility or clarity:**
> >
> > 1. For Figure 2, we make the legend bigger, labels clearer, and color more friendly. Besides, we also improve its captions.
> > 2. Regarding Figure 3, the x-axis represents the update step, and its units were already provided in the original submission. Given that each sub-figure contains four lines, and considering the error bars might compromise readability, we have additionally provided the ablation study with error analysis during optimization in the Appendix A9 and following table to better demonstrate convergence behavior and stability. As illustrated in the following table and Figure 12 in the revised submission, we demonstrate the optimization trajectory and stability of convergence across three different random seeds and data subsets. **It can be seen that, the variance of the optimization process across different subsets is small, and for different seeds on the same subset, the variance is negligible.**
> >
> >
> > Table: Optimized values and their variance across three different seeds with DiSF values.
> > | Steps | 0 | 2000 | 4000 | 6000 | 8000 | 10000 |
> > | :--- | :--- | :--- | :--- | :--- | :--- | :--- |
> > | Random Seed 1 | -0.213893 | -0.176177 | -0.168775 | -0.165291 | -0.163269 | -0.162287 |
> > | Random Seed 2 | -0.213867 | -0.176057 | -0.168761 | -0.16528 | -0.163267 | -0.162276 |
> > | Random Seed 3 | -0.213844 | -0.176042 | -0.168714 | -0.165223 | -0.163258 | -0.162268 |
> > | Mean | -0.213868 | -0.176092 | -0.168750 | -0.165265 | -0.163265 | -0.162277 |
> > | Std. | 0.000020 | 0.000060 | 0.000026 | 0.000030 | 0.000005 | 0.000008 |
> > | Variance | 0.009150 | 0.006206 | 0.005697 | 0.005464 | 0.005331 | 0.005267 |
> >
> > Table: Optimized values and their variance across three different subsets with Pair-wise Similarity.
> > | Steps | 0 | 2000 | 4000 | 6000 | 8000 | 10000 |
> > | :--- | :--- | :--- | :--- | :--- | :--- | :--- |
> > | Random Subset 1 | -0.11915 | -0.115665 | -0.115005 | -0.114674 | -0.114494 | -0.114444 |
> > | Random Subset 2 | -0.118995 | -0.115601 | -0.11494 | -0.114609 | -0.114431 | -0.114378 |
> > | Random Subset 3 | -0.119114 | -0.115555 | -0.114852 | -0.114523 | -0.114337 | -0.114296 |
> > | Mean | -0.119086 | -0.115607 | -0.114932 | -0.114602 | -0.114421 | -0.114373 |
> > | Std. | 0.000066 | 0.000045 | 0.000063 | 0.000062 | 0.000065 | 0.000061 |
> > | Variance | 0.002839 | 0.002675 | 0.002645 | 0.002630 | 0.002621 | 0.002619 |

---

> ### Author Response · Authors · 2025-11-21
> **Rebuttal to Reviewer 5FG7 (3/3)**
>
> > **W3 and Q2:** Missing Comparison with Scalable Learned Selection Paradigms Comparisons are made primarily against heuristic baselines. A comparison with other learned selection methods that are scalable to trillion-token datasets (if available) is needed to better situate DATAMASK's advantages. For methods limited to small-scale data, this could be framed as future work instead of a current limitation. 2. It seems valuable to compare DATAMASK with learned data selection strategies that are scalable to trillion-token datasets. For methods limited to small-scale data, could you elaborate on why they are not suitable for direct comparison, or outline how
>
> **Reply to W3 and Q2:**
>
> To the best of our knowledge, existing approaches for learning-based diversity maximization on set functions can be generally divided into efficient implementation (Semdedup), classical greedy algorithms (DiSF), and reinforcement or model based methods (DVRL, DPPNet, and FLEXSUBNET recommanded by another reviewer). **In the original submission, we compared Semdedup and DiSF. Therefore, in the following, we compare DVRL, DPPNet, and FLEXSUBNET with our DATAMASK.**
>
> **As compared in the following Table T1, DVRL, DPPNet, and FLEXSUBNET utilize a small validation set or submodular functions to train a model to predict sampling probabilities for a given batch of data in a small data scale. In sharp contrast, our method directly optimizes the mask sampling probabilities for all samples based on any specified objective function in large scale of data.** To facilitate the comparison, we follow the DVRL pipeline to train a Data Value Estimator, which consists of three linear modules with ReLU activation, followed by a final sigmoid activation. It was trained on 1,000,000 text samples, each with 768 dimensions of features, using a reward based on Pair-wise Similarity (averaged cosine similarity). After convergence, the Data Value Estimator predicted the logits for these 1,000,000 samples as well as for another 1,000,000 unseen samples. Based on these predictions, we selected the top 10% of samples and recorded their exact pairwise similarity compared with our DATAMASK framework. As shown in the following table T2, we present the pairwise similarity and the time taken to train the model until full convergence. **During our implementation of DVRL with large-scale pre-training text data, they show the following issues:**
> - **Sub-optimal performance on the seen set.** Unlike our DATAMASK framework, which directly optimizes the sampling probabilities of all samples based on the metric, DVRL trains a model to predict the probabilities for a given batch of data. This approach amplifies the errors arising from the model's learning capacity, achieving much worse converged values after optimization.
> - **Poor generalization ability on the unseen set and huge computation.** We observe significantly larger scores on the unseen set compared to the seen set, revealing a lack of generalization in DVRL. This implies that despite DVRL training a model to predict the probabilities, the model must be re-trained when handling new data. Given that a single DVRL model already requires significantly more computation (6x) than our DATAMASK, the need for repeated retraining renders DVRL computationally infeasible for selecting large-scale pre-training data.
> - **Unstable training.** Following DVRL's pipeline, we identified significant divergence risks at the 1M sample scale. After careful selection of hyperparameters and engineering efforts , we finally achieved relatively satisfactory optimized values.
>
> **We hope these enrich the comparison of our DATAMASK to Learned Selection Paradigms, and help address the reviewer's concern.**
>
> | Table T1| Data | Data scale | Trained Model | Model scale | Method | computation |
> | :--- | :--- | :--- | :--- | :--- | :--- | :--- |
> | DVRL | Cifar100 | 0.06M | ResNet-32 | <1 M | Training the data value estimator using a reinforcement signal of the reward obtained on a small validation set that reflects performance on the target task. | 8 hour on 0.06M samples |
> | DPPNet | MNIST, Celeb | 0.06M | Wide-ResNet | <1 M | Develop an attention mechanism based on the transformer that captures a notion of dissimilarity between feature vectors | - |
> | FLEXSUBNET | Amazon baby registry dataset | 0.1M | neural network | <1 M | Flexible neural models for both monotone and non-monotone submodular functions for selection. | - |
> | Ours | FineWeb | 15,000M | 1/7 B LLM | 1-7,000 M (1-7B) | Mask learning with any function without the need of validation set or additional model | 40 minutes on 1M samples |
>
> | Table T2| **Before optimization** | **Seen Set (after optimization)** | **Unseen Set (after optimization)** | **Computational time** |
> | :--- | :--- | :--- | :--- | :--- |
> | DVRL | 0.7101 | 0.6874 | 0.7040 | 2h30min |
> | DATAMASK | 0.7101 | 0.6831 | - | 40min |

---

### Author Response · Authors · 2025-11-21
**General Response**

**We would like to thank all the reviewers(5FG7, iTj3, nve6 and NCCX) for their thoughtful suggestions on our paper, and appreciate that the reviewers have multiple positive impressions of our work, including:**
- **Well-motivated** (Reviewers iTj3, nve6) and **novel** (Reviewers 5FG7, nve6).
- **Well-structured and clearly written** (Reviewer 5FG7), supported by **rigorous empirical validation** (Reviewers 5FG7, nve6) and a **transparent cost breakdown** (Reviewer iTj3).
- **Strong results**, indicating higher quality in selected samples (Reviewer NCCX) and significant improvements across various tasks (Reviewer 5FG7).
- **High engineering scalability and practicality** (Reviewers nve6, NCCX), demonstrating **significant value** for LLM pre-training (Reviewer 5FG7) and data selection (Reviewers 5FG7, iTj3).

**We provide a summary of our responses in the following, with most corresponding discussions included in the revised submission, highlighted in blue color for clarity. For detailed responses, please refer to the feedback for each comment/question point-by-point.**

**About the revised submission:**
- We promise to release the code for the data curation recipe as well as the FineWeb-Mask dataset (for reviewer iTj3).
- We have revised the manuscript based on the reviewers' constructive comments and added approximately 4 pages to the Appendix to incorporate the new discussions and ablation studies (for all reviewers).

**Introduction:**
- We improved Figure 2 with bigger legends, clearer labels clearer, and more friendly colors. Besides, we also improve its captions (for reviewer 5FG7).

**Rethinking and Related Works:**
- We added a new research work GneissWeb and provided a primary comparison to its methodology (for reviewer iTj3).
- We compare our approach with learned selection paradigms and explain why they may not scale well with large-scale LLM pretraining data. Specifically, we implement DVRL on 1M samples, compared to our DATAMASK (for reviewers 5FG7, nve6).
- We highlighted the baseline DCLM that is already included (for reviewer nve6).

**Method:**
- We provided a theoretical analysis and outlined how to ensure stability during optimization (for reviewer nve6).
- We better clarify words like "optimized score", "group number G"(for the reviewer iTj3), and explain independence between different rollout (for reviewer NCCX).
- We explained the issues of unstable estimation and the change in the selection target when directly assigning logit values based on each sample's quality and diversity scores (for reviewer NCCX).
- We demonstrate that our union optimization framework can extend to more than two metrics, specifically by adding the text toxicity metric as requested (for reviewer nve6).

**Experiments and Appendix:**
- We extended our ablations to include the learning rate (for reviewers 5FG7 and NCCX), updating epochs, and initialization strategy for logits (for reviewer 5FG7).
- We improved Figure 3 and provided more detailed explanations and ablations to interpret convergence behavior and stability (for reviewers 5FG7, NCCX).
- We extended experimental settings and showed the effectiveness of our method during stage-two pretraining following post-training (for reviewer iTj3).
- We provided additional visualizations of the selected samples' distribution across domains and text lengths, compared to the baselines (for reviewer nve6).

**We appreciate all reviewers’ time and effort again. We have included most corresponding discussions, reviewer-recommended related work and experimental results into the revised submission. We are looking forward to your reply!**

---

### Meta-Review · Area_Chair_BDmM · 2026-01-05

**Summary:**

Overall, reviewers were positive about the work and viewed it as a strong and timely contribution to large-scale LLM pre-training data curation. The strengths outlined the novelty of formulating joint quality–diversity data selection as a policy-gradient-based mask learning problem, the scalability of DATAMASK to trillion-token datasets, and the empirical validation conducted across multiple model sizes and downstream tasks (Reviewers 5FG7, iTj3, and NCCX). The release of the FineWeb-Mask dataset and the transparent cost analysis were also highlighted as valuable contributions to the community (Reviewers 5FG7 and iTj3).

In my view, the primary concerns raised by reviewers focused on methodological clarity, robustness, and positioning rather than major issues with the work. These included limited ablation of certain hyperparameters (e.g., learning rate, initialization), clarity and accessibility issues in some figures and explanations, and questions about the necessity of policy-gradient optimization versus simpler alternatives (Reviewers 5FG7 and NCCX). Moreover, questions were raised about the applicability of the selected dataset size for full pre-training versus stage-two pre-training settings (Reviewer iTj3). In sum, reviewers considered these issues addressable and largely framed them as opportunities to further strengthen the impact of the paper.

**Reviewer Concerns:**

The rebuttal and revised submission addressed the majority of reviewer concerns in a thorough and convincing manner. In particular, requests for broader and more systematic ablations were directly addressed by adding analyses on learning rate, updating epochs, and logit initialization. The authors also strengthened comparisons to related work by adding discussion and experiments contrasting DATAMASK with scalable learned selection paradigms (e.g., DVRL at smaller scale) and by incorporating recent related datasets such as GneissWeb. Several clarity/presentation issues were also resolved satisfactorily.

**Reviewer Scores:**

- Reviewer 5FG7: I believe they would have likely maintained their score or increased it to an 8 given concerns were addressed and no new ones emerged.

- Reviewer iTj3: I believe they would have increased their score to an 8 since concerns were clearly addressed as they mentioned in their response to authors.

- Reviewer NCCX: They would have maintained their score at 6 as they indicated to authors.

- Reviewer nve6: Would have increased their score to 6 at least.

---

### Decision · Program_Chairs · 2026-01-26

Accept (Poster)